



# Updates to the Met Office's global ocean-sea ice forecasting system including model and data assimilation changes

Davi Mignac[1], Jennifer Waters[1], Daniel J. Lea[1], Matthew J. Martin[1], James While[1], Anthony T. Weaver[2], Arthur Vidard[3], Catherine Guiavarc'h[1], Dave Storkey[1], David Ford[1], Edward W. Blockley[1], Jonathan Baker[1], Keith Haines[4], Martin R. Price[1], Michael J. Bell[1], and Richard Renshaw[1]

[1]Met Office, Exeter, United Kingdom
[2]CERFACS, Toulouse, France
[3]Inria, Grenoble, France
[4]University of Reading, Reading, United Kingdom

*Correspondence to*: Davi Mignac (davi.carneiro@metoffice.gov.uk)

**Abstract.**

The Forecast Ocean Assimilation Model (FOAM) is the Met Office's operational, coupled ocean-sea ice system, which produces analyses and short-range forecasts at global and regional scales each day for various stakeholders, including defence, marine navigation and science users. This paper describes and evaluates the impacts of recent model and data assimilation (DA) updates on global FOAM when compared to its current operational version. The model updates include the use of the TEOS10 formulation for the seawater equation of state, with improved ocean model settings in the Southern Ocean and the implementation of a new sea ice model. Updates to the DA include an increase in the number of DA minimisation iterations, an improved specification of observation errors for sea surface temperature and sea level anomaly (SLA), and optimisations of the DA computational efficiency. Large-scale DA corrections for temperature have also been removed to prevent an inconsistent projection of the SLA DA signal onto large-scale temperature at depth. For one-year runs at 1/12° resolution, the new FOAM system shows a 40% improvement in observation-minus-background (OmB) statistics for SLA and sub-surface temperatures relative to the current system in eddy-rich regions, which result in a similar level of improvement for ocean currents. To evaluate potential impacts on the pre-Argo period, one-year experiments at 1/4° resolution are run withholding profiles of temperature and salinity observations in both new and current FOAM systems. In the absence of profile DA, OmB statistics for SLA, temperature and salinity in the new FOAM system can reach improvements up to 90% in the Southern Hemisphere relative to the current system, resulting in more temporally consistent ocean transport and heat content results. Therefore, it is expected that the model and DA updates will lead to more potential for use of FOAM reanalyses in climate studies, particularly in the pre-Argo period, and will provide improved ocean/sea-ice initial conditions to FOAM as well as to the Met Office short-range and seasonal coupled ocean/atmosphere/land/sea ice forecasting systems.



## 1 Introduction

Ocean forecasting systems are a key component of Operational Oceanography, defined as a set of activities for the generation of products and services providing regular information on the marine environment (Davidson et al., 2019). Global ocean and sea ice monitoring, forecasting and reanalysis systems require continuous implementation of the most advanced research findings to meet user needs and provide information that is crucial for the research community to gain major understanding and advanced knowledge in the marine sector (Skákala et al., 2024). In this context, the Forecast Ocean Assimilation Model (FOAM; Barbosa-Aguiar et al., 2024) is the Met Office's operational, coupled ocean-sea ice system, producing analyses and short-range forecasts of ocean and sea ice at global and regional scales each day. FOAM contributes to the OceanPredict international collaboration for developing ocean forecasting capability, in conjunction with systems from various other operational centres (see recent reviews by Davidson et al., 2019 and Moore et al., 2019). As part of OceanPredict, FOAM has been included in global ocean forecast intercomparison studies, showing a competitive performance against velocity drifter observations (Aijaz et al., 2023) and the best accuracy for mid-latitude temperatures and salinities (Ryan et al., 2015) when compared to other global systems. Global FOAM forecasts are primarily produced for use by the Royal Navy, but they are also used for other applications involving safety at sea and shipping, offshore operations and monitoring of oil spills and pollutants (Davidson et al., 2009; Jacobs et al., 2009; Brushett et al., 2011; Davidson et al., 2019; Bilge et al., 2022). The global FOAM system also provides boundary conditions for regional forecasting systems, such as the FOAM North-West European shelf system (Tonani et al., 2019).

FOAM has been run operationally since 1997 and has been continually evolving as an ocean forecasting system since then (Bell et al., 2000; Blockley et al., 2014; Waters et al., 2015; Barbosa-Aguiar et al., 2024). It currently uses the Nucleus for European Modelling of the Ocean (NEMO; Madec et al., 2022) community ocean model coupled to the Los Alamos sea ice model (CICE; Hunke and Lipscomb, 2015). An important aspect of FOAM and the other Met Office forecasting systems, such as the coupled Numerical Weather Prediction (NWP; Guiavarc'h et al., 2019) and seasonal forecasting systems (GloSea; MacLachlan et al., 2014), is their initialisation. The ocean and sea ice components of all these different systems are initialised using a common data assimilation (DA) framework to support the Met Office's aim of producing seamless forecasts across all timescales (Brown et al., 2012). This ocean-sea ice DA framework is called NEMOVAR and consists of an incremental, multi-variate, multi-length scale 3DVar scheme (Waters et al., 2015; Mirouze et al., 2016). NEMOVAR operationally assimilates observations from varied sources: swath satellite and in situ sea-surface temperature (SST) observations, in situ temperature (T) and salinity (S) profiles, along-track sea-level anomaly (SLA) from altimeter observations and satellite sea ice concentration (SIC) data.

Upgrades to model versions of the global FOAM, the coupled NWP and the GloSea seasonal forecasting systems are coordinated so that the initial conditions are appropriate for the forecast model used operationally. The GloSea system also requires a long reanalysis of the ocean to initialise the reforecasts needed for real-time forecast calibration, and the system used to produce this reanalysis needs to be consistent with the real-time forecasting system. The model development process



in the Met Office is organised through so-called Global Coupled model versions which consist of specific versions of the
Global Atmosphere (GA), Land (GL), Ocean (GO) and Sea Ice (GSI) model configurations. The global FOAM currently uses
the Global Ocean configuration version 6 (GO6; Storkey et al., 2018) coupled to the Global Sea Ice configuration version 8.1
(GSI8.1; Ridley et al., 2018), which were implemented in September 2018. Updated versions of the GO and GSI components
(GOSI9) have been developed since then, including changing from CICE to the new NEMO sea-ice model SI$^3$ (Sea Ice
modelling Integrated Initiative; Vancoppenolle et al., 2023).

Alongside the model developments, improvements to the ocean/sea-ice DA capability are a continuous process. This includes
development and implementation of underpinning algorithmic improvements to NEMOVAR (Lea et al., 2008; Mirouze et al.,
2016; Weaver et al., 2016; While and Martin, 2019; Weaver et al., 2020), inclusion and testing of new observation types (Lea
et al., 2014; Martin et al., 2019; Mignac et al., 2022; Waters et al., 2024; King et al., 2024), as well as improvements to the
error covariances and other ancillary input information needed by DA (Waters et al., 2015; Barbosa-Aguiar et al., 2024).
Changes to the ocean/sea-ice DA are thoroughly tested in a research and development framework first. Once these changes
are properly validated, they are then implemented as part of the operational system, often as a package with a new model
version.

This paper describes and evaluates the impact of model and DA changes implemented in GOSI9 on the performance of global
FOAM at 1/4° and 1/12° horizontal resolutions when compared to the current operational version GO6+GSI8.1 (hereafter GO6
for simplicity), as described by Barbosa-Aguiar et al. (2024). The paper is outlined as follows. Section 2 presents an overview
of the model changes in GOSI9. Section 3 focuses on giving a description of the DA changes implemented at the same time
as upgrading to GOSI9. Section 4 presents a comparison between FOAM GOSI9 and GO6 by looking at the following
assessments: the impacts of the GOSI9 changes on the global FOAM configuration at 1/12º horizontal resolution in one-year
experiments; and potential impacts of the GOSI9 changes on FOAM reanalyses before the Argo period by running another set
of one-year experiments with the 1/4° system withholding T/S profile data. Finally, Section 5 summarises the impact of the
model and DA changes and draws some conclusions.

## 2 GOSI9 model updates

### 2.1 Ocean model changes

Two global FOAM configurations are used here, one at 1/4° horizontal resolution (ORCA025) and one at 1/12° horizontal
resolution (ORCA12). These are both run using a tripolar horizontal grid where the two poles in the northern hemisphere are
placed over land to avoid singularities in the part of the grid where computations are carried out. Both configurations have 75
vertical levels, with about 1 m vertical resolution in the top 10 m of the ocean. As well as the model grid and its vertical levels,
ORCA12 and ORCA025 bathymetries in the GOSI9 model version are unchanged from GO6. See Barbosa-Aguiar et al. (2024)
for a more detailed description of the global FOAM configurations at GO6.





The version of the NEMO base code has been upgraded from NEMO 3.6 in GO6 to NEMO 4.0.4 in GOSI9. A new implicit-adaptive vertical advection scheme is implemented in NEMO 4.0.4 (Shchepetkin, 2015). This allows implicit vertical advection to be used in regions where the vertical Courant-Friedrichs-Lewy (CFL) condition is likely to be breached, while maintaining an explicit vertical advection scheme elsewhere. This development keeps the accuracy of the explicit scheme implemented in NEMO 3.6 but allows for a longer time-step. Therefore, the time-step for FOAM ORCA12 has increased from

180 seconds in GO6 to 400 seconds in GOSI9, and this produces a significant improvement in runtime from approximately 50 minutes to 26 minutes for a 24-hour ORCA12 run. A 4th order tracer advection scheme is also implemented in GOSI9, as opposed to the 2nd order scheme used in GO6, which was found to reduce spurious vertical mixing (Guiavarc'h et al., 2024). There was also tuning of the turbulent kinetic energy mixing depth between 10°S and 40°S in GOSI9, improving warm subsurface biases in the Indian Ocean (Guiavarc'h et al., 2024). Additionally, there was further tuning to increase the Antarctic

Circumpolar Current (ACC) transport in GOSI9, including an increased topographic drag in the Southern Ocean due to changing the lateral boundary condition from a free-slip to a partial-slip condition south of 50°S (Storkey et al., 2024). In GOSI9, the effects of unresolved eddies at high latitudes are accounted for in both ORCA025 and ORCA12 by introducing a space- and time-dependent version of the Gent-McWilliams scheme (Tréguier et al., 1997). It follows the approach of Hallberg (2013) that eddies should be explicitly represented in parts of the domain where the model resolution is sufficiently fine and

parameterised where it is not. Therefore, in the ORCA12 system the Gent-McWilliams scheme is only applied at very high latitudes (i.e., poleward of ~60°S and 60°N).

A significant development for GOSI9 is the change of the equation of state from EOS80 to TEOS10. TEOS10 was adopted as the official description of seawater by the Intergovernmental Oceanographic Commission in 2009 (IOC, 2010). The TEOS10 equation of state changes the temperature and salinity variables used in the model and analysis from potential temperature and

practical salinity to conservative temperature and absolute salinity, respectively. The largest differences between potential temperature and conservative temperature are near the surface and in enclosed seas, whereas the difference of practical salinity minus absolute salinity is negative everywhere (see Fig. S1). A new processing step has been included in NEMO 4.0.4 to allow the conversion of in situ temperature and practical salinity observations to conservative temperature and absolute salinity prior to assimilation, respectively. Since a corresponding salinity value is required for the temperature conversion, when there is a

corresponding salinity observation (e.g., for Argo floats), the observed salinity is used in the conversion, otherwise the model salinity is used. This is the same approach used in GO6 for the conversion from in situ to potential temperature.

## 2.2 New sea ice model

The sea ice model component of GOSI9 is based upon the native NEMO sea ice model SI[3], which was developed from the

Louvain-La-Neuve sea ice model version 3 (LIM3) with some functionality merged from CICE. SI[3] is fully embedded in the NEMO code. The version of SI[3] used at GOSI9 is based on NEMO 4.0.4. Aside from the change in the sea ice model, the sea-ice physics remains largely similar to the GSI8.1 CICE configurations used with GO6. Like CICE, SI[3] is a dynamic-thermodynamic continuum sea-ice model that includes an ice thickness distribution, conservation of horizontal momentum,





an elastic-viscous-plastic rheology, and energy-conserving halo-thermodynamics. As in CICE, five thickness categories are

used to model the sub-grid-scale ice thickness distribution in $SI^3$, and an additional ice-free category represents open water. $SI^3$ is run on the same grid as the NEMO ocean model component, with fields exchanged between the ocean and sea ice on every time-step. An advantage of using the ice model native to NEMO is that the interpolation of velocity points required between NEMO (Arakawa C-grid) and CICE (Arakawa B-grid) in previous configurations is no longer necessary. For a more detailed description of the sea ice model in GOSI9, see Blockley et al. (2024). Table 1 also summarises the model changes

between GOSI9 and GO6.

## 3 GOSI9 data assimilation updates

### 3.1 NEMOVAR changes

In this section, we address the individual impact of each GOSI9 DA update on FOAM, whereas the results showing the overall impacts due to all GOSI9 changes are reserved for Section 4. In FOAM, the NEMOVAR system is used to perform multi-

variate, incremental 3DVar with first-guess-at-appropriate-time. Model values are interpolated to the observation locations at the nearest model time-step during a one-day model forecast. The observation-minus-model values (called the innovations) are fed into the NEMOVAR code which generates a set of changes (called the analysis increments) to bring the model closer to the observations. These increments are then added during the analysis model run uniformly over one day, using an Incremental Analysis Update (IAU; Bloom et al., 1996) scheme. An important feature of NEMOVAR is the balance operator,

which allows covariances between different ocean variables to be accounted for. The balance operator is represented by physically-based balance relationships, and variables used in the 3DVar scheme are transformed to a set of assumed mutually uncorrelated control variables, namely temperature, unbalanced salinity and unbalanced sea surface height (SSH). After generating the analysis increments, these variables are transformed back to the state variables of temperature, salinity, SSH and the horizontal velocities (see Waters et al., 2015). No balance relationships are defined between the ocean and SIC

variables, whose increments are obtained from a separate NEMOVAR minimisation. It is worth mentioning that ORCA12 DA is currently performed on the ORCA025 grid since it is very computationally expensive to run the DA at the full ORCA12 resolution. A recent description of the way NEMOVAR is implemented in FOAM is included in Barbosa-Aguiar et al. (2024). In GO6, total observation error variances are specified through seasonally and spatially varying estimates produced using the method of Hollingsworth and Lönnberg (1986). The way the SLA and SST observation errors are estimated has changed in

GOSI9, with total observation errors being calculated for these variables through the combination of measurement and representation errors (REs) separately. The method of Oke and Sakov (2008) is applied in GOSI9 to estimate the SLA and SST REs, by accounting for the model sub-grid scale variability present in the observations. Provided that there are at least four observations, which adequately span the model grid cell, the RE is then computed as the standard deviation of the differences between each observation and the observation average within each grid cell for a particular point in time. If there

are less than four observations within a grid cell, the RE calculation is not performed and the specific grid cell is masked. SLA



REs are calculated every 5 days for 2017-2018, using along-track data from all the available altimeters within a 5-day window, to mitigate sampling errors. For the SST, only night-time data from the Visible Infrared Imaging Radiometer Suite (VIIRS) instrument are used. This is because VIIRS is the reference dataset for the bias correction of SST observations in FOAM (While and Martin, 2019), and it has good daily global coverage. The SST REs are calculated every 3 days for 2017-2018

using a 3-day window. Both SST and SLA REs are averaged over the seasons to be used as input files for NEMOVAR. After averaging the REs, additional steps are done, such as smoothing the fields and infilling any ocean grid cell where the minimum number of observations, required for the RE calculation, is not met. Since ORCA12 DA is currently performed at ORCA025 resolution, the same SST and SLA REs for ORCA025 are used in GOSI9 ORCA12 DA.

On top of the seasonally averaged REs, a measurement error of 4 cm is included for SLA observations, resulting in the SLA

observation errors used by NEMOVAR in GOSI9. SLA observation errors in GOSI9 are much larger than those in GO6, which are mostly smaller than the measurement error of 4 cm used in GOSI9 (Fig. 1). This suggests that the SLA assimilation in GO6 may have been overfitting the observations. In contrast to SLA, SST measurement errors are defined for each single observation, using the measurement uncertainties given by the data providers. The use of observation-specific SST measurement error in a global FOAM configuration is a new development in GOSI9. SST measurement errors in GO6 are

only represented by seasonal estimates, making them more generic and less optimal for DA (see Fig. S2).

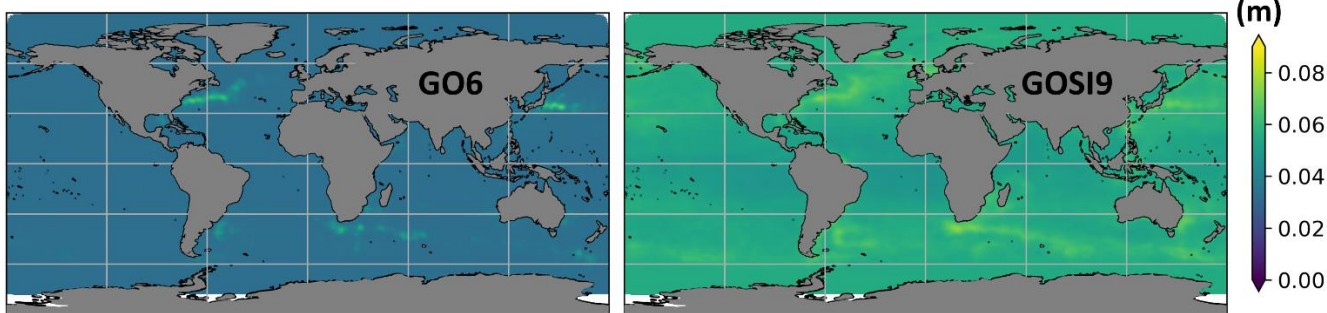

**Figure 1. ORCA025 SLA observation errors (m) for the December to February season for GO6 (left) and GOSI9 (right).**

In the FOAM implementation of NEMOVAR, the background error covariances for each control variable are parametrised by

a combination of error variances and horizontal and vertical error correlation length-scales. Spatially and seasonally varying background error variances at the surface are specified, with a parametrisation based on the vertical temperature gradients in the background field determining the error variances below the surface (Waters et al., 2015). Vertical error correlation length-scales are then specified based on the background mixed-layer depth. The vertical length-scales and variances of the background errors have not been changed between GO6 and GOSI9. The horizontal error correlations are specified based on

the combination of two length-scales (Mirouze et al., 2016) and the way these are used has changed in GOSI9. In GO6, the small-scale background error covariances have a length-scale which depends on the first baroclinic Rossby radius, while the large-scale error covariances have a 400 km length-scale for temperature, unbalanced salinity and unbalanced SSH. A major





update made in GOSI9 is that only the short length-scales of the background errors are used to horizontally spread the temperature information. The large-scale DA corrections for temperature are removed in GOSI9 to mitigate problems with the projection of the SLA signal onto large-scale temperature at depth, which can lead to drifts and spurious variability in sub-surface temperatures and heat content, particularly in the pre-Argo period (see Fig. S3).

Removing the large-scale DA temperature corrections results in improved observation-minus-background (OmB) temperature statistics, which show a decrease of the Root Mean Squared Difference (RMSD) relative to T profile observations (Fig. 2a-c), particularly in the South Pacific and Southern Ocean (see Fig. 3 for the limits of each ocean basin). However, the long length-scales of the background errors for the unbalanced salinity are retained to prevent near-surface drifts (Fig. 2e-f), due to the sparsity of salinity observations. The unbalanced salinity is not used in the density calculation that is fed to the dynamic height relationship for computing the SSH balance in NEMOVAR. Therefore, the SLA signal does not project onto unbalanced salinity, which allows us to keep the long length-scale for salinity background errors.

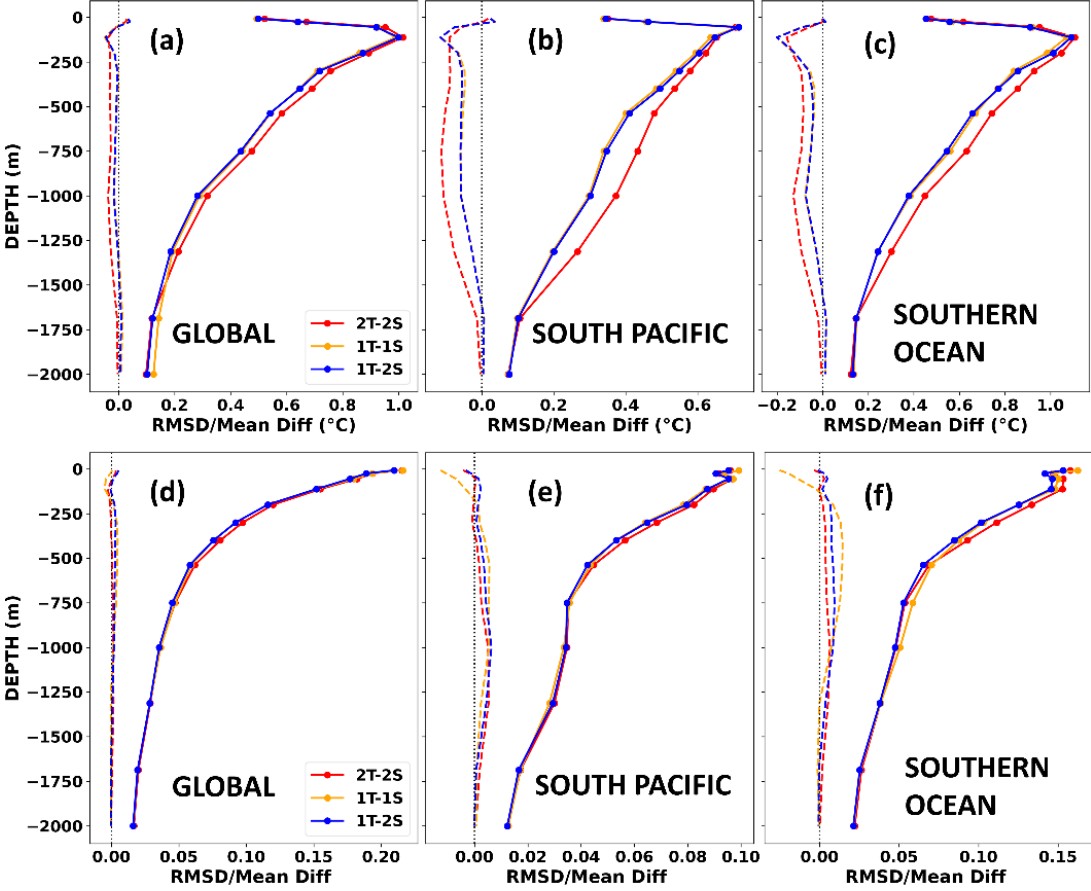

**Figure 2. ORCA025 OmB RMSD (solid) and mean difference (dashed) for (a-c) temperature (°C) and (d-f) salinity of FOAM runs using different length-scale setups for T/S background errors. The red line is the run using both short and long length-scales for T and S (2T-2S), the orange line is the run using only short length-scales for T and S (1T-1S), and the blue line is the run using only short length-scales for T but both short and long length-scales for S (1T-2S). OmB statistics are calculated against T/S profile observations for January-May 2019.**




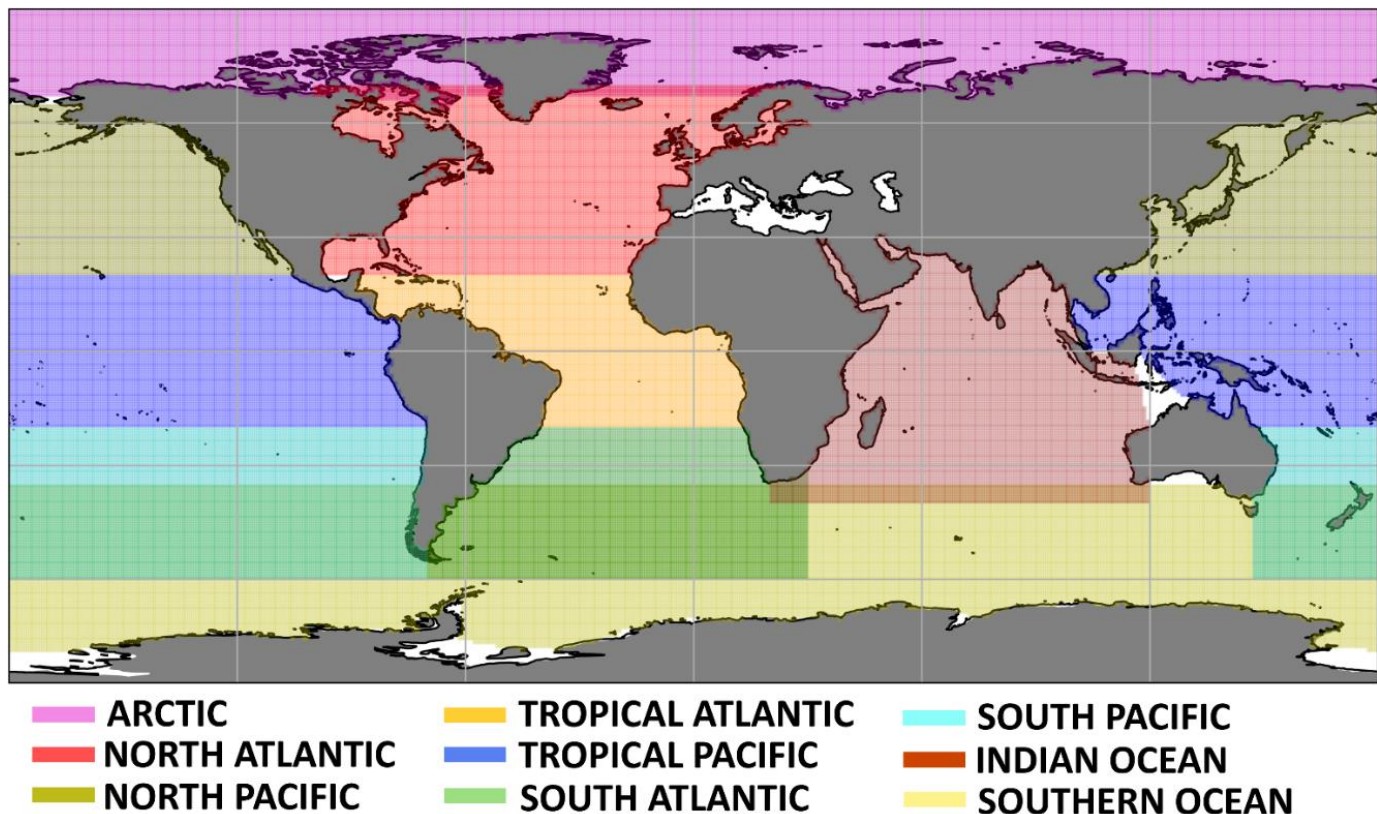

**Figure 3. Areas of the different ocean basins used to calculate regional OmB statistics. Note that there are intersections between the ocean basins, particularly between the Southern Ocean and its neighbouring basins, such as the South Pacific and South Atlantic.**

The NEMOVAR SSH balance is now applied throughout the whole water column in GOSI9, whereas in GO6 the SSH balance is applied only below the mixed layer, i.e. only density changes below the mixed layer are used to generate balanced SSH increments (and vice versa). This change reduces water column instabilities caused by DA in areas of water mass convection, such as the Mediterranean Outflow region, resulting in improved SLA statistics in those areas, particularly for ORCA12 (see Fig. S4). Although the Mean Dynamic Topography (MDT) product used in the SLA assimilation is going to be updated from

CNES-CLS13 (Rio et al., 2014) to a more recent version in GOSI9, the impacts of using different MDT products were still being assessed when the final experiments of this paper were run. Therefore, the assessment of using a more recent MDT product in the SLA assimilation is beyond the scope of this paper. The equation of state in the NEMOVAR balance operator has also been updated to use TEOS10.

The 40 inner loop iterations in NEMOVAR, as configured in GO6, were not always enough to guarantee sufficient convergence

of the minimisation of the 3DVar cost function, which is done iteratively using a B-preconditioned conjugate gradient algorithm (Gürol et al., 2014). Therefore, the number of inner loop iterations has increased to 120 in GOSI9 for the ocean DA





component, resulting in SLA RMSD improvements across the different ocean basins, particularly in the western boundary currents and the ACC (see Fig. S5). Since SIC is analysed independently from the other ocean variables, the sea-ice DA component kept the number of 40 inner loop iterations as in GO6.

Although the sea-ice model changed from CICE to SI$^3$ in GOSI9, the way the total SIC increments are added to the sea ice model remains the same. Positive ice concentration increments are always added to the thinnest ice category (consisting of ice up to 0.45 m), while negative increments are first removed from the thinnest available category until it reaches zero concentration, and then progressively removed from thicker categories, as in Peterson et al. (2015). All other prognostic variables in SI$^3$, such as the ice and snow volume, snow and ice enthalpy, melt pond fraction and volume, are adjusted

proportionally to changes in SIC, so that equivalent values can be maintained after adding SIC increments. Adjusting melt pond variables, due to SIC DA changes, is also a GOSI9 improvement relative to GO6. See Tab. 1 for a summary of changes between GOSI9 and GO6 DA.

### 3.2 Brunt-Väisälä verifications for T/S increments

Although the DA changes in Section 3.1 resulted in major improvements (see Section 4), the fact that a long length-scale is used in GOSI9 to horizontally spread the unbalanced salinity but not the temperature information, may exacerbate GO6 water column instabilities in very sensitive regions, such as in deep convection locations. Changes were made in the IAU, so that Brunt-Väisälä buoyancy frequencies ($N^2$) are now computed using salinity and temperature fields after their respective increments are added on each time-step of the IAU. T/S increments for the whole water column are then rejected at each grid

point if any $N^2$ value indicates water column instability (i.e., $N^2 < 0$) within a specified depth range. In order to target deep convection areas, the depth range for the Brunt-Väisälä checks is set to be between 400 and 1500 m globally, except for the Mediterranean Outflow region, where the initial depth is chosen to be 150 m. In the Mediterranean Outflow region, a shallower depth range is chosen to better consider water column instabilities starting at the base of the mixed layer.

   Positive impacts in FOAM GOSI9 can be clearly seen in areas of water mass convection. For instance, noisy patches of colder

water appearing at depth in the Labrador Sea, which result from deep convection being triggered by DA-induced water column instabilities, are largely mitigated with the $N^2$ verifications (Fig. 4a-c). This is consistent with OmB improvements found in the Labrador Sea, particularly for the 250-2000 m mean differences and 1000-2000 m RMSDs relative to T profile observations, when FOAM GOSI9 applies $N^2$ constraints to T/S increments (Fig. 4d). Similarly, the use of $N^2$ values to reject T/S increments prevents shallow water instabilities from occurring in the Mediterranean Outflow and improves T/S statistics

for FOAM ORCA12 in this region (see Fig. S6). Given these positive impacts, we plan to further develop our current Brunt-Väisälä verification scheme in the future. For instance, the vertical profile of T/S increments can be iteratively adjusted on each IAU time-step to always ensure consistent $N^2$ values. This will avoid situations with the current scheme where all the vertical profile of T/S increments may be neglected at a grid point location because of a single negative $N^2$ value at depth.








**Figure 4. FOAM ORCA025 temperature fields (°C) at 1200 m in the Labrador Sea for (a) GO6, (b) GOSI9 and (c) GOSI9 additionally applying Brunt-Väisälä verifications on the T/S increments. These are snapshots of 31 March 2019 after all runs have been initialised on 1 January 2019 from the same initial condition. (d) FOAM ORCA025 OmB RMSD (solid) and mean difference (dashed) relative to T profiles observations between January and April 2019 for the Labrador Sea region.**

## 3.3 Optimisations to the DA computational efficiency

The background error correlations in NEMOVAR are modelled using an implicit diffusion operator (Weaver et al., 2016). The resulting matrix must be normalised to ensure that the diagonal elements are approximately equal to one. These normalisation factors are expensive to calculate, yet if the correlation length-scales vary from one assimilation cycle to the next, they need



to be re-calculated. In practice, this makes it very expensive to use flow dependent correlation length-scales in NEMOVAR. The normalisation factor look-up table, containing a set of discrete mixed layer depths, is a pragmatic solution to allow flow-dependent vertical correlation length-scales in FOAM (Waters et al., 2015). The vertical length-scales depend on the local mixed layer depth while the horizontal length-scales are fixed. For a particular cycle of NEMOVAR, the mixed layer depths

are calculated from the model fields for that day and then at each horizontal location a profile of normalisation factors is extracted from the field in the look-up table with the corresponding mixed layer depth (Waters et al., 2015). However, it is still very expensive to perform the initial calculation of the look-up table, particularly for ORCA12, and it uses a significant amount of input/output during the running of NEMOVAR. Weaver et al. (2020) proposed an alternative approach where the calculation of the horizontal and vertical normalisation factors can be separated, which we refer to as "decomposed normalisation factors".

Although calculating the 3D field of normalisation factors for the horizontal correlations is still done offline, this new approach allows us to obtain a new 3D field of the exact vertical normalisation factors for the flow-dependent vertical scales on each assimilation cycle. The decomposed normalisation factor approach is computationally affordable and does not impact the quality of GOSI9 DA results, while reducing the input/output requirements used in NEMOVAR when compared to the look-up table.

In addition to the decomposed normalisation factors, a multi-resolution assimilation using the transfer grid functionality (TRF) of NEMOVAR is implemented for the first time in FOAM at GOSI9. The implicit diffusion calculation used for generating the spatial correlation functions, which are employed in the background covariance model, is one of the most expensive operations in NEMOVAR. The TRF option allows this to be run on a coarser grid. The disadvantage is that the correlation functions are smoother, but this is acceptable for the longer background error length-scales which are many multiples of the

grid size even in ORCA025. A two times coarsening option is applied in GOSI9 for the long background error length-scales, representing a 50% reduction in the NEMOVAR runtime without significantly changing the DA results. This runtime reduction is particularly important in GOSI9 because it compensates for the NEMOVAR runtime increase due to tripling the number of inner loop iterations (see Section 3.1). Therefore, it takes 13 minutes using 15 computational nodes to perform the DA at ORCA025 resolution in both FOAM GO6 and GOSI9, even though FOAM GOSI9 has tripled the number of DA iterations.




|  | **FOAM GO6** | **FOAM GOS19** |
|---|---|---|
| **Ocean model** | NEMO 3.6 | NEMO 4.0.4 with improved parameter settings in the Southern Ocean (Storkey et al., 2024), 4th order tracer advection scheme, and space- and time-dependent version of Gent-McWilliams scheme |
| **Sea ice model** | CICE5 | $SI^3$ 4.0.4 |
| **Equation of state** | EOS80 | TEOS10 |
| **SST and SLA observation errors** | Seasonally and spatially varying estimates produced using the Hollingsworth and Lönnberg method. This is treated as the total error. | Seasonally and spatially varying representation error due to unresolved scales in the model (Oke and Sakov, 2008) + observation-specific measurement error for SST and 4 cm measurement error for SLA |
| **Background error correlation length-scales** | Short and long length-scales are used for both T and S | Long length-scale is not applied for T but is used for S |
| **Inner loop iterations (ocean DA)** | 40 | 120 |
| **SSH balance** | Applied below, but not in the mixed layer | Applied through the whole water column |
| **Rejection of T/S increments** | None | Rejection of T/S increments based on water column instabilities diagnosed from Brunt-Väisälä buoyancy frequencies |
| **Normalisation factors** | Look-up table | Decomposed normalisation factors |
| **Multi-resolution for implicit diffusion model** | Not implemented | Implemented for ORCA025 DA, which is the grid resolution that is also used by ORCA12 DA. |

**Table 1. Summary of model and DA differences between FOAM GO6 and GOS19 systems.**



## 4 FOAM GOSI9 vs FOAM GO6

### 4.1 Experiment setup

Table 2 shows one-year runs that have been conducted for 2019 to compare the performance of FOAM GOSI9 and GO6 under different circumstances. These runs have been done for both ORCA12 and ORCA025, assimilating all the observation types as in the operational configuration (see Tab. 1 in Barbosa-Aguiar et al., 2024). We first assess the impacts of the GOSI9 changes on the global FOAM configuration at 1/12º (see Section 4.2). We then present FOAM GO6 and GOSI9 runs withholding all T/S profile observations for ORCA025 only. This allows us to evaluate GOSI9 potential impacts on FOAM reanalyses at 1/4º in the pre-Argo period (see Section 4.3), which is known to have heat content drifts and spurious variability in GO6.

Surface forcing is provided by the operational Met Office global atmospheric model's archived output in the form of hourly fields of 10-m winds together with three-hourly fields of 10-m temperature, 10-m specific humidity, precipitation and radiation fluxes at ~10 km horizontal resolution. Apart from conversions between EOS80 and TEOS10 and between CICE and SI$^3$ variables, the initial condition for both FOAM GO6 and GOSI9 runs are the same. They come from 2017-2018 ORCA12 and ORCA025 FOAM runs with the GO6 configuration (see Barbosa-Aguiar et al., 2024).

| | Experiment description | Configuration |
|---|---|---|
| **GO6** | FOAM GO6 run including all the observation types assimilated operationally | ORCA025/ORCA12 |
| **GO6-NoTSProf** | Same as FOAM GO6 but withholding T/S profile observations | ORCA025 |
| **GOSI9** | FOAM GOSI9 run including all the observation types assimilated operationally | ORCA025/ORCA12 |
| **GOSI9-NoTSProf** | Same as FOAM GOSI9 but withholding T/S profile observations | ORCA025 |

**Table 2. Configuration of the FOAM GO6 and GOSI9 experiments.**

OmB statistics are calculated against T/S profile observations (including Argo, XBTs, CTDs and other sources), along-track SLA from all available altimeters, SST from in situ drifters and SIC from the Special Sensor Microwave Imager/Sounder (SSMIS). Although these observations are compared to the model background, i.e. before being assimilated, they cannot be treated as fully independent datasets. Therefore, to further enhance our OmB assessments, we also look at comparing FOAM GO6 and GOSI9 with respect to independent observation datasets, such as the meridional (V) and zonal (U) velocities at 15 m depth from in situ drifter positions. These are produced by the Copernicus Marine Service (doi: 10.17882/86236) and are not currently assimilated in FOAM. Daily mean model velocities at 15 m depth are interpolated to the location of the drifter-derived velocities, with a rotation applied to convert from the model grid reference frame to the V and U directions of the observed velocities. The observations and model equivalents are then used to calculate the OmB statistics.

To better quantify FOAM GOSI9 impacts relative to GO6 for each variable, the OmB RMSD for both FOAM systems are compared using the equation below:



$$\alpha = \left( \frac{RMSD_{GOSI9} - RMSD_{GO6}}{RMSD_{GO6}} \right) * 100 \qquad (1)$$

where negative (positive) α corresponds to FOAM GOSI9 RMSD improvements (degradations) in percentage relative to GO6.

It is also worth noting that the temperature and salinity RMSD results for FOAM GO6 and GOSI9 are calculated from the EOS80 and TEOS10 variables, respectively. The magnitude of the errors is expected to be consistent whether using TEOS10 or EOS80. We investigated the impact of converting between absolute and practical salinity on the OmB values and found that it has a very small impact of the order of 0.001, which is much smaller than the salinity differences and RMSDs between FOAM GO6 and GOSI9 presented here.


## 4.2 Impacts on the global FOAM system with all observation types assimilated

ORCA12 SLA statistics are clearly improved in FOAM GOSI9 compared to GO6, showing a global OmB RMSD decrease from 0.067 to 0.058 m, while the mean differences remain largely unaffected and close to zero relative to the SLA observations (Fig 5a). RMSD improvements are more significant from April 2019 onwards, indicating that there is a spin-up period of ~4

months for FOAM ORCA12 SLAs to adjust to the new configurations in GOSI9. Spatially, RMSD improvements in FOAM GOSI9 can easily reach up to 40% relative to GO6, especially in the western boundary current regions and along the ACC path (Fig. 5c). Although smaller than in the western boundary current regions, FOAM GOSI9 RMSD improvements are noted everywhere for SLA, except for the central and eastern North Atlantic region, a few coastal areas, and enclosed seas.

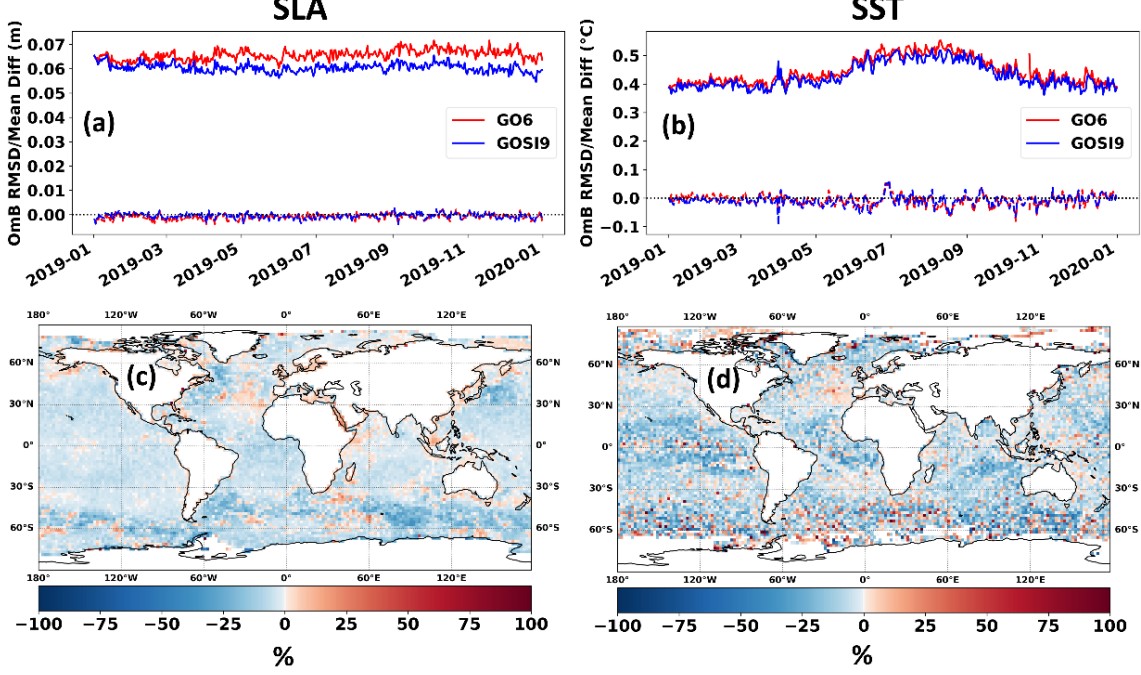

**Figure 5. (a, b) 2019 ORCA12 OmB statistics for FOAM GO6 (red) and GOSI9 (blue), calculated against SLA (m) and SST (°C) observations, respectively. The solid lines correspond to RMSDs, and the dashed lines represent mean differences. (c, d) 2019 RMSD percentage improvements (blue) and degradations (red) of FOAM ORCA12 GOSI9 relative to GO6 for SLA and SST, respectively. Observations used for SLA are all the along-track altimeters available, whereas for SST observations used are in situ drifters.**





Globally, FOAM GOSI9 slightly improves the OmB SST statistics when compared to GO6, showing a RMSD decrease from

0.46 to 0.44°C (Fig. 5b). These SST improvements are consistent spatially, particularly in the tropical regions, where a ~30% RMSD improvement in FOAM GOSI9 can be found across all ocean basins relative to GO6 (Fig. 5d). However, it is worth highlighting that small RMSD degradations are seen for the FOAM GOSI9 SSTs in the Mediterranean Outflow region and in the Arctic. Although removing large-scale DA temperature corrections can lead to consistent FOAM improvements, it has already been mentioned in Section 3.2 that a drawback of this change is that it can exacerbate localised water column

instabilities in regions very sensitive to T/S increments. Small improvements in FOAM GOSI9 SSTs are also consistently seen when compared to surface temperature measurements from profile observations (see Fig. 6).

GOSI9 changes in FOAM have also led to a decrease in the global RMSDs and biases of the model temperatures at depth when compared to GO6, particularly between 250 and 1500 m, relative to profile observations (Fig. 6). These global improvements are largely driven by the substantial impacts of the GOSI9 changes in the Southern Hemisphere, particularly in

the South Atlantic, South Pacific and Southern Ocean. For instance, temperature RMSDs at depth are nearly halved in FOAM GOSI9 for these ocean basins. Despite the largest impacts being in the Southern Hemisphere, improvements in the OmB temperature statistics are also clearly noted in the North Atlantic and North Pacific. This reinforces that removing large-scale DA corrections for temperature leads to improvements in how the SLA observation information projects onto model temperatures at depth in FOAM GOSI9 (see also Fig. 2).

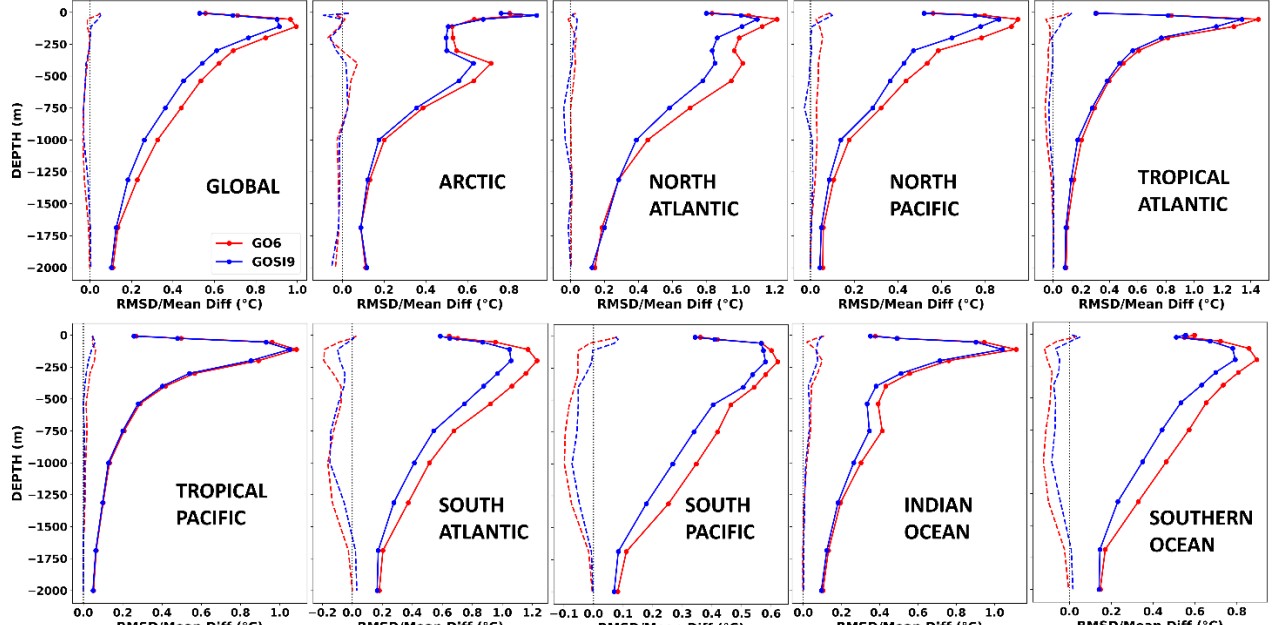


**Figure 6. 2019 ORCA12 OmB temperature statistics (°C) for FOAM GO6 (red) and GOSI9 (blue) calculated against profile observations in different ocean basins. The solid lines correspond to RMSDs, and the dashed lines represent mean differences.**

When compared to GO6, FOAM GOSI9 salinity improvements also occur at depth in the same regions where temperature improvements are seen, such as in the South Pacific, South Atlantic, Southern Ocean, North Atlantic and North Pacific (Fig.





7). This indicates that removing the large-scale DA corrections for temperature also has an indirect effect on improving the sub-surface salinity structure in both hemispheres. Although a long length-scale is still used to propagate the unbalanced salinity increments, small near-surface drifts are still noted in FOAM GOSI9, particularly in the South Pacific and Southern Ocean. These near-surface salinity drifts may be avoided if a model bias correction scheme for salinity is used within the assimilation (e.g., Balmaseda et al., 2007), which looks like a promising next step to further improve FOAM GOSI9 results.

An additional FOAM GOSI9 run was performed with only the model changes but leaving the DA configuration unchanged from GO6. The purpose of this additional run was to better evaluate the relative contribution of the model and DA changes to FOAM GOSI9 improvements. The model changes alone only have a small impact on the OmB statistics (see Fig. S7-S9), except for the SLA in the Southern Ocean, where the model changes contribute to ~37.5% of the total GOSI9 RMSD improvement in this region. Therefore, the DA changes (see Section 3) explain most of the FOAM GOSI9 RMSD decrease in

SLA, SST and T/S with respect to GO6.

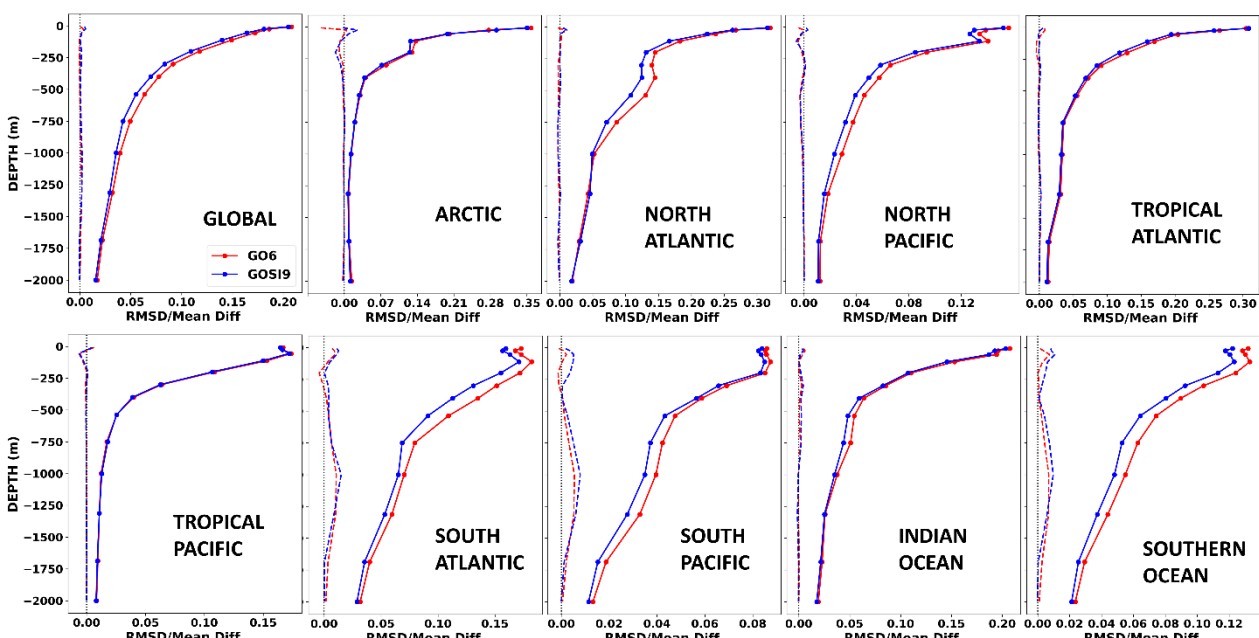

**Figure 7. 2019 ORCA12 OmB salinity statistics for FOAM GO6 (red) and GOSI9 (blue) calculated against profile observations in different ocean basins. The solid lines correspond to RMSDs, and the dashed lines represent mean differences.**

Despite the consistent FOAM GOSI9 improvements with respect to GO6, all the OmB statistics so far have considered

observation types that are assimilated in FOAM. Hence, we look at comparing FOAM GO6 and GOSI9 with drifter-derived velocities (Fig. 8), which can be treated as an independent dataset. Like the other ocean variables, FOAM GOSI9 shows a consistent improvement in the OmB statistics relative to GO6, particularly in the RMSD, for both V (Fig. 8a) and U (Fig. 8b) velocity components over time. This also translates to better RMSD statistics spatially (Fig. 8c-f), where FOAM GOSI9 improvements are clearly noted for both V and U within the western boundary currents, such as the Gulf Steam and Kuroshio

current, as well as along the ACC path. In fact, V and U RMSD improvements in FOAM GOSI9 can reach up to ~40% in



these regions when compared to GO6 (Fig. 8g-h). As one would expect, due to the strong physical links between SLA and ocean currents, these GOSI9 velocity enhancements in FOAM are very similar, both spatially and in RMSD improvement percentage, to GOSI9 SLA enhancements (Fig. 5c). However, U and V degradations are seen in FOAM GOSI9 along the equator. No balanced velocity increments are applied near the equator in FOAM, so the model velocities are not properly

constrained by DA in this region.

**Figure 8. (a, b) 2019 ORCA12 OmB statistics for FOAM GO6 (red) and GOSI9 (blue), calculated against V and U velocity components (m/s) at 15 m from drifter observations, respectively. The solid lines correspond to RMSDs, and the dashed lines represent mean differences. 2019 (c, d) FOAM GO6 and (e, f) GOSI9 RMSDs with respect to drifter observations for V and U, respectively. (g, h) 2019 RMSD percentage improvements (blue) and degradations (red) of FOAM ORCA12 GOSI9 relative to GO6 for V and U, respectively.**




The ORCA12 SIC performance in FOAM GOSI9 is mixed when compared to GO6 (Fig. 9). There are clear SIC improvements in the Arctic summer, with both RMSDs and mean differences decreasing from ~0.06 to ~0.04 and from ~0.015 to almost zero, respectively (Fig. 9a). These SIC summer improvements in FOAM GOSI9 are even more significant in ORCA025 (see Fig. S10). Spatially, ORCA12 RMSD SIC improvements in FOAM GOSI9 are shown in the Arctic ice pack over summer, reaching up to 50% in the North of Greenland and in the Canadian Archipelago with respect to GO6 (Fig. 9e). FOAM GO6 is known to show excessive sea ice melting over the Arctic summer, due to feedback issues between the DA and the melt ponds (Barbosa-Aguiar et al., 2024). For instance, the model will melt sea ice and form ponds over summer, then DA adds new sea ice but does not reverse the ponding, which leads to more sea ice melting because the pond albedo is considerably lower than the ice albedo. As mentioned before, when SIC increments are added to $SI^3$, the SIC DA changes are proportionally propagated to more prognostic variables than in CICE, including the melt pond variables. This might explain why there is less sea ice melting in FOAM GOSI9 over the Arctic summer.

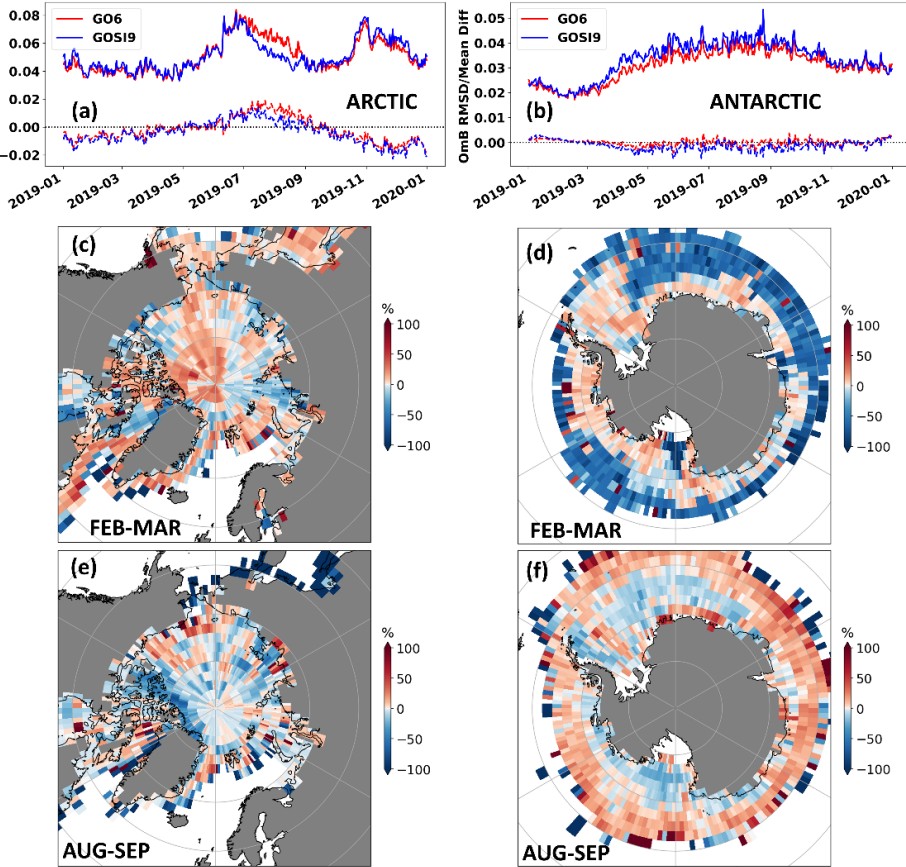

**Figure 9. 2019 ORCA12 OmB statistics for FOAM GO6 (red) and GOSI9 (blue), calculated against SSMIS SIC observations in the (a) Arctic and (b) Antarctic. The solid lines correspond to RMSDs, and the dashed lines represent mean differences. (c-d) February-March and (e-f) August-September RMSD percentage improvements (blue) and degradations (red) of FOAM ORCA12 GOSI9 relative to GO6 for the Arctic and Antarctic, respectively.**



Despite the improvements in the Arctic summer, FOAM GOSI9 SIC results are slightly worse than in GO6 in other Arctic seasons (Fig. 9a), as well as throughout the year in the Antarctic (Fig. 9b). As shown in Fig. 9c-f, SIC RMSD degradations in FOAM GOSI9 are more dominant in the winter for both Arctic and Antarctic, showing a notable contrast to the summer season, in which results are mostly improved in the Arctic but mixed in the Antarctic. We note that small absolute SIC changes could result in large percentage changes in areas where SIC is very low (i.e., near the ice edge). Hence, although FOAM GOSI9

RMSD percentage improvements relative to GO6 are clearly large near the ice edge in the Antarctic summer (Fig. 9d), the overall SIC RMSD in FOAM GOSI9 is still slightly worse than in GO6 for the same period in the Antarctic (Fig. 9b). The SIC background error covariances in GOSI9 are still derived from a GO6 run, even though they have different sea ice models. Additionally, the same $SI^3$ settings from ORCA025 are applied to ORCA12, which suggests that there is room for further ORCA12 SIC improvements from both DA and modelling perspectives.


### 4.3 Potential impacts on the global FOAM pre-Argo reanalysis

In this section, we assess the potential impacts of the GOSI9 changes on ORCA025 FOAM reanalysis before the Argo period, by running another set of 2019 experiments withholding T/S profile observations (see Tab. 2). The advantage of this experimental set-up is that we can compare results to the withheld T/S profiles to indicate the likely impact the changes will

have before Argo data are available for either assimilation or assessment. A downside of the set-up is that we only run for a one-year period, so any longer time-scale drifts will not be so apparent.

As shown in Fig. 10a, the global OmB SLA statistics in GOSI9-NoTSProf are much improved when compared to GO6-NoTSProf. While SLA RMSDs increase in GO6-NoTSProf throughout 2019, showing a clear degradation in the OmB SLA statistics without the T/S profile assimilation, the GOSI9-NoTSProf run holds very similar SLA RMSDs in comparison to its

original FOAM GOSI9 run. We see that GOSI9-NoTSProf performs even better than the FOAM GO6 run assimilating T/S profiles. For the global SST statistics (Fig. 10b), the GOSI9 impacts on FOAM are less clear, although small improvements are still seen in GOSI9-NoTSProf compared to GO6-NoTSProf throughout the year.

Most of the SLA and SST RMSD degradations in GO6-NoTSProf develop quickly at high latitudes, particularly in the Southern Ocean, where the SLAs and SSTs can reach up to a 100% RMSD degradation along the ACC path relative to its FOAM GO6

original run (Fig. 10c-d). This is not the case for GOSI9-NoTSProf, which mostly maintains the positive SLA and SST impacts seen in the FOAM GOSI9 run assimilating T/S profiles (Fig. 10e-h), including the Southern Ocean. These results are very promising for pre-Argo ocean reanalysis runs in GOSI9, since both SST and SLA OmB statistics in FOAM GOSI9 are not significantly degraded in the absence of T/S profile assimilation. However, it is worth highlighting that FOAM SLA statistics in the central and eastern North Atlantic are slightly better in GOSI9-NoTSProf (Fig. 10g) than in GOSI9 (Fig. 10e). This

suggests that there could still be minor issues in assimilating SLA and T/S profile data together, even after the substantial SLA improvements caused by GOSI9 DA changes.



In the absence of T/S profile assimilation, FOAM GO6 configuration significantly degrades the sub-surface temperatures and salinities, showing very large RMSDs and mean differences across the different ocean basins, particularly in the South Pacific, South Atlantic and Southern Ocean (Fig. 11 and Fig. 12). Although there is an expected degradation in the GOSI9-NoTSProf

experiment, it is much smaller than in GO6-NoTSProf, with temperature and salinity RMSDs at depth decreased by a factor of 3 globally when these two runs are compared. Likewise, the global mean differences relative to T/S profile observations are much closer to zero in GOSI9-NoTSProf relative to GO6-NoTSProf, especially between 250 and 1500 m. This just reinforces that removing the large-scale DA corrections of temperature in FOAM GOSI9 significantly mitigates the degradation of sub-surface temperatures and salinities. This is a known FOAM GO6 issue caused by an inconsistent large-scale propagation of

the SLA DA signal onto model temperatures at depth, which is exacerbated when there is no T/S profile assimilation.

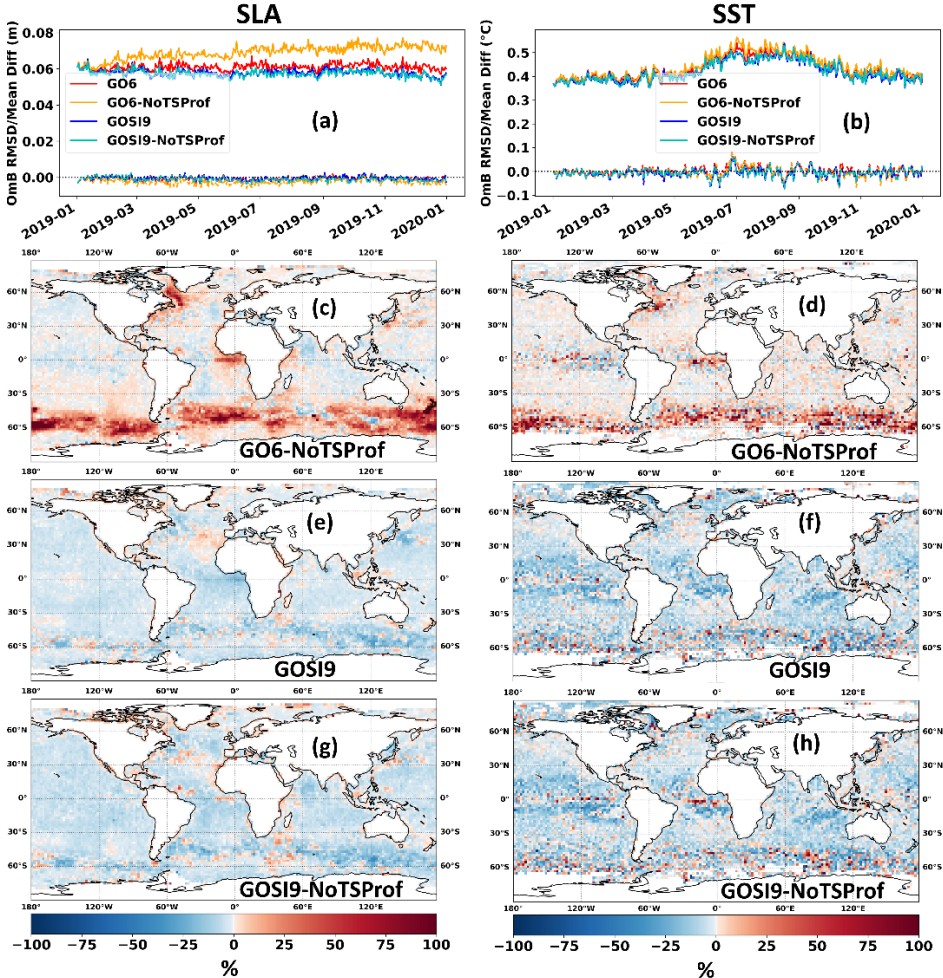

**Figure 10. (a, b) 2019 ORCA025 OmB statistics for FOAM GO6 (red), GO6-NoTSProf (orange), GOSI9 (blue) and GOSI9-NoTSProf (cyan), calculated against SLA (m) and SST (°C) observations, respectively. The solid lines correspond to RMSDs, and the dashed lines represent mean differences. 2019 RMSD percentage improvements (blue) and degradations (red) of (c, d) GO6-NoTSProf, (e,**

**f) GOSI9 and (g, h) GOSI9-NoTSProf relative to GO6 for SLA and SST, respectively. Observations used for SLA are all the along-track altimeters available, whereas for SST observations used are in situ drifters.**




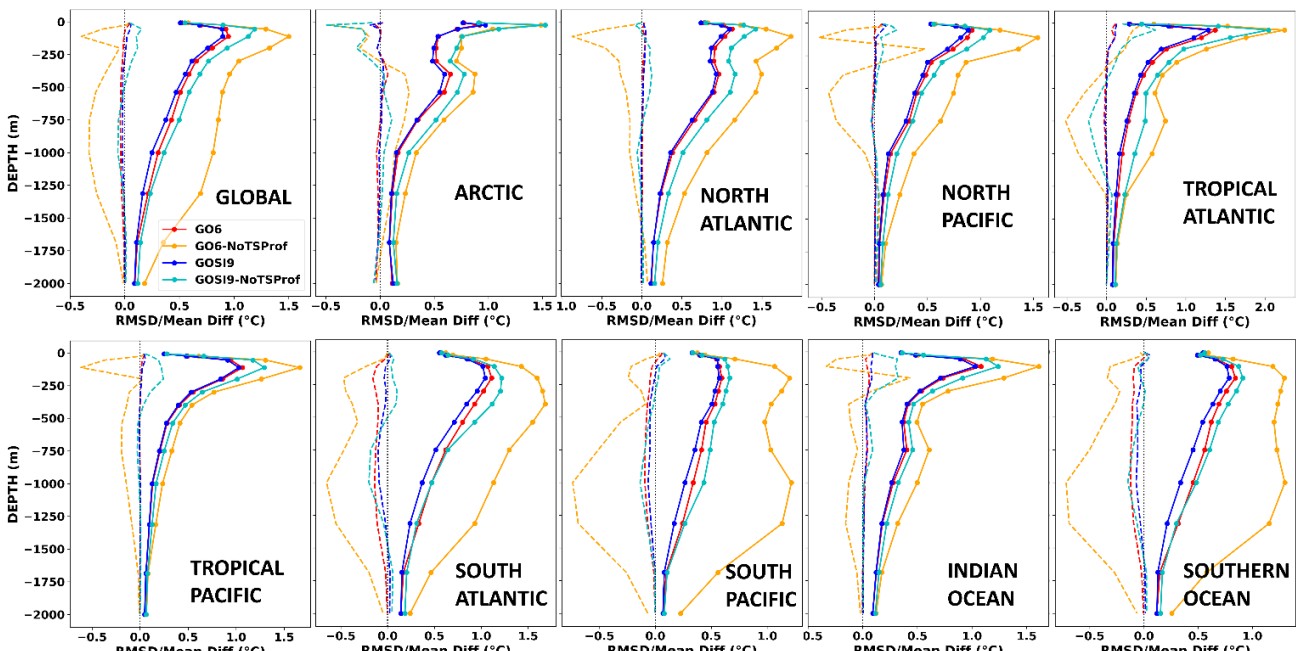

**Figure 11. 2019 ORCA025 OmB temperature statistics (°C) for FOAM GO6 (red), GO6-NoTSProf (orange), GOSI9 (blue) and GOSI9-NoTSProf (cyan) calculated against profile observations in different ocean basins. The solid lines correspond to RMSDs, and the dashed lines represent mean differences.**


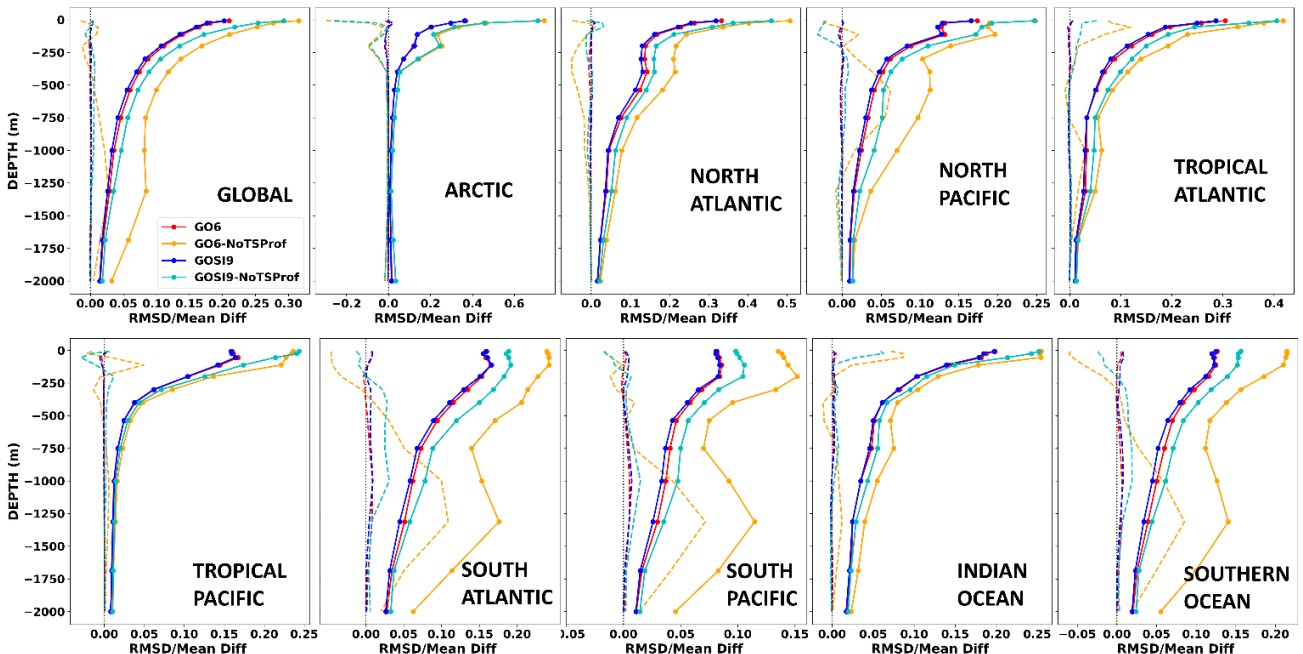

**Figure 12. 2019 ORCA025 OmB salinity statistics for FOAM GO6 (red), GO6-NoTSProf (orange), GOSI9 (blue) and GOSI9-NoTSProf (cyan) calculated against profile observations in different ocean basins. The solid lines correspond to RMSDs, and the dashed lines represent mean differences.**





In addition to the OmB statistics presented so far, we also include here some additional diagnostics looking at the impacts of the GOSI9 changes on the Atlantic Meridional Overturning Circulation (AMOC) and heat content, which are important key indicators from a reanalysis perspective. For Fig. 13 and Fig. 14, a 2019 run turning off the assimilation of all observation types (NOASSIM) is added, so we can compare the magnitude of FOAM GO6 and GOSI9 drifts in the heat content and ocean transports with NOASSIM.

The heat content time series for the global ocean shows that GOSI9-NoTSProf drifts much less than GO6-NoTSProf when both are compared to EN4 monthly objective analysis and their original FOAM runs assimilating T/S profile observations (Fig. 13). The drifts in GO6-NoTSProf are so large between 700 and 2000 m that its heat content between these depths lies completely outside the range of EN4 uncertainties. This is consistent with the fact that these are the depths where the largest RMSD temperature degradations occur in GO6-NoTSProf relative to the other runs (Fig. 11). On the other hand, 700-2000 m

heat content drifts are largely mitigated in GOSI9-NoTSProf, which follows very closely GOSI9 and EN4 heat contents. However, although much smaller than in GO6-NoTSProf, heat content drifts are still present in GOSI9-NoTSProf for depths between 0 and 700 m. As one would expect, some heat content drift is likely to be present in the pre-Argo period due to the lack of T/S profile observations. This is supported by the fact that GOSI9-NoTSProf run drifts very closely to the NOASSIM run between 0 and 700 m. Future work will involve running longer FOAM GOSI9 reanalyses to look at these 0-700 m heat

content drifts in more detail.

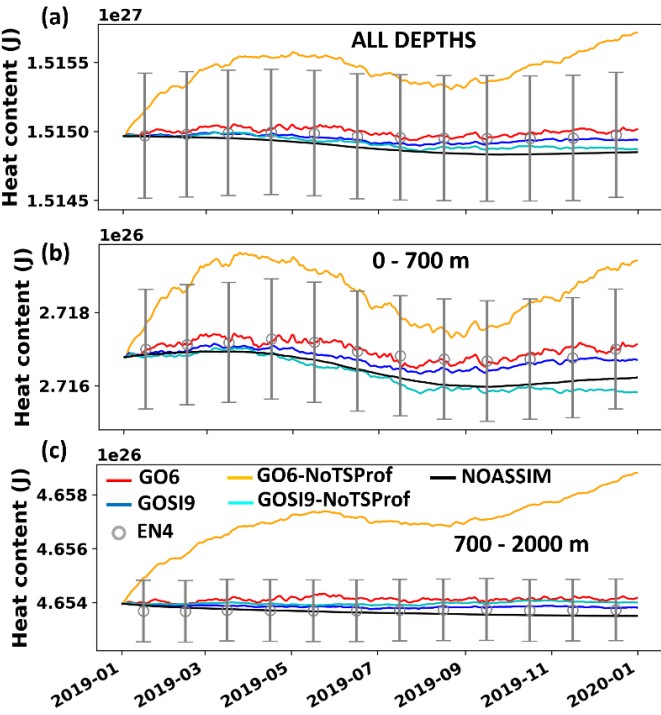

**Figure 13. Daily heat content (J) time series for the global ocean calculated (a) at all depths, (b) 0-700 m and (c) 700-2000 m. Monthly heat content and its uncertainties from the EN4 objective analysis are also shown. The heat content uncertainties are derived from EN4 monthly temperature uncertainties.**




Although FOAM runs assimilating T/S profiles agree relatively well with the EN4 heat content, FOAM GO6 shows much noisier daily heat content variability than FOAM GOSI9, particularly between 0 and 700 m (Fig. 13). This noise also seems to be a consequence of the large-scale signal from SLA assimilation being inconsistently propagated onto the model temperatures at depth in GO6, and this was noted by Dong et al. (2021) when seeking to apply a DA smoother to FOAM GO6 results. Since these large-scale SLA corrections onto sub-surface temperatures are not applied anymore in GOSI9, the daily heat content variability is smoother and should now enable a consistent application of the DA smoother, as in Dong et al. (2021), to further improve FOAM GOSI9 reanalysis results.

As well as for the heat content, the GOSI9 DA impacts on the AMOC transports are positive (Fig. 14). The monthly AMOC transports in GOSI9-NoTSProf follow very closely those from the FOAM GOSI9 run assimilating T/S profiles, particularly in the Northern Hemisphere, and still show relatively good agreement with RAPID transports at 26.5°N (Fig. 14a). On the other hand, the AMOC in GO6-NoTSProf significantly drifts, producing increased and unrealistic transports at 50°N and 26.5°N (Fig. 14a-b), which do not agree at all with the RAPID transports and with any of the other runs. Although the drifts in the AMOC are largely reduced in GOSI9-NoTSProf in the Northern Hemisphere, it is worth noting that differences of ~4 Sv occur between this run and the FOAM GOSI9 run assimilating T/S profiles at 30°S, before converging again to similar transports in December 2019 (Fig. 14c). Longer AMOC time series in the Southern Hemisphere will be evaluated in future FOAM GOSI9 reanalyses, even though previous results have shown that T/S, SST and SLA OmB statistics are clearly improved in the Southern Hemisphere with GOSI9 DA changes. Therefore, it is expected that the ocean circulation will also respond positively to these DA changes in the Southern Hemisphere.

Similar FOAM GOSI9 results can also be found when we look at the AMOC stream function (Fig. 14d-h). In the absence of T/S profile assimilation, GOSI9-NoTSProf AMOC stream function is not as intense as in its equivalent GO6-NoTSProf run in the Northern Hemisphere, but still stronger than NOASSIM just north of the equator. The upper AMOC transports are significantly weakened south of the equator in GO6-NoTSProf, particularly between 10°S and the equator, and this is also significantly improved in GOSI9-NoTSProf where the AMOC is more meridionally coherent across the Atlantic, particularly in the equatorial region.

**5 Summary and Conclusions**

GOSI9 updates to the Met Office's global Forecast Ocean Assimilation Model (FOAM) have consistently shown positive impacts on the configuration at 1/12º (ORCA12) and 1/4º (ORCA025). The impacts on global FOAM were evaluated for 2019 by comparing GOSI9 against GO6, which is the current version used by the operational global FOAM system. Further FOAM experiments were also conducted to test the likely impact of GOSI9 changes on the pre-Argo period reanalysis by running further GO6 and GOSI9 ORCA025 experiments in 2019 withholding T/S profile observations.




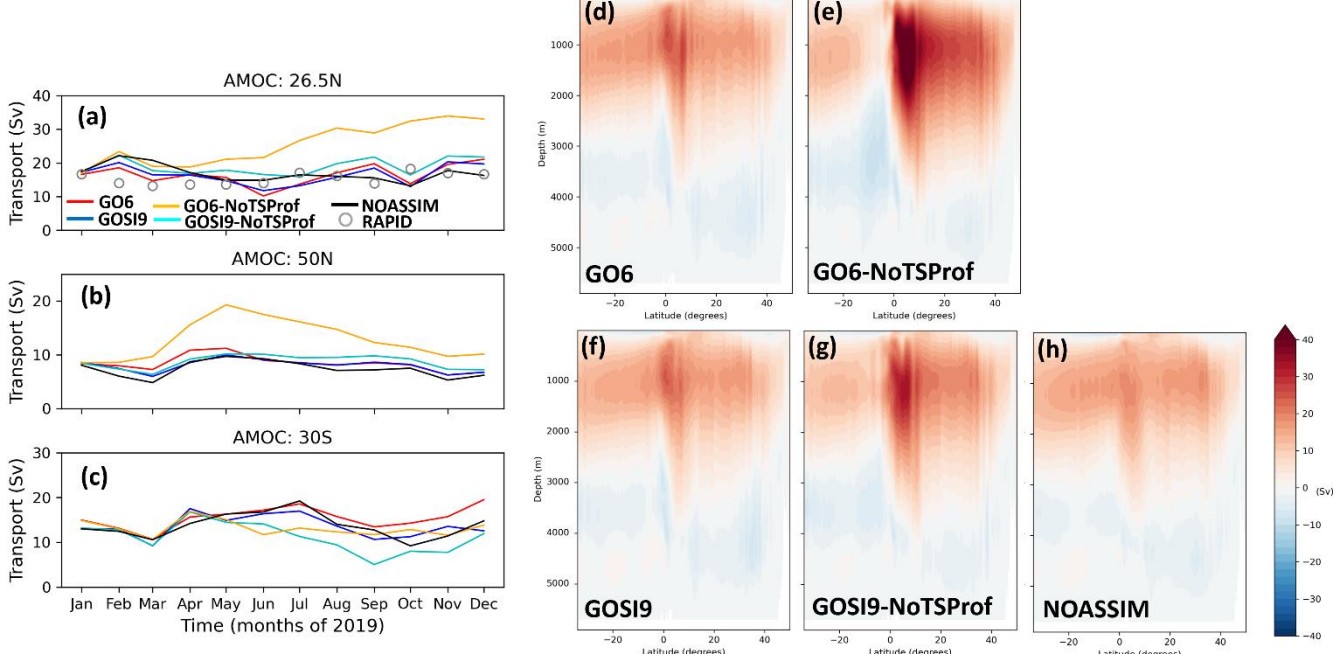

**Figure 14. Monthly AMOC transports (Sv) in 2019 at (a) 26.5°N, (b) 50°N and (c) 30°S for FOAM GO6 (red), GO6-NoTSProf (orange), GOSI9 (blue), GOSI9-NoTSProf (cyan) and NOASSIM (black). Note that observed AMOC transports from the RAPID array (grey) are also included at 26.5°N. (d-h) AMOC stream function (Sv) for all the FOAM runs.**

Although GOSI9 consists of both model and DA changes, the latter were the main contributor to improving FOAM GOSI9 observation-minus-background (OmB) statistics with respect to GO6. Major model changes were implemented to improve the Southern Ocean in climate simulations, but they are designed to work on longer timescales, i.e. from months to years (Storkey et al., 2024) and so only had a small impact on the short-range forecasts assessed here. Therefore, the DA changes, such as the increased number of inner loop iterations, improved SST and SLA observation error specifications, and particularly the removal of large-scale DA corrections for model temperatures at depth, led to a much better ORCA12 performance in FOAM GOSI9 relative to GO6. SLAs and sub-surface temperatures showed a substantial decrease in global OmB RMSDs and mean differences in FOAM GOSI9 when compared to GO6. The GOSI9 DA changes also produce slightly better FOAM results for sub-surface salinities, although near-surface salinity biases are marginally increased in the Southern Ocean with respect to GO6.

The SLA and sub-surface temperature improvements in FOAM GOSI9 ORCA12 are largely driven by enhanced OmB statistics in the South Atlantic, South Pacific and Southern Ocean, although consistent temperature and SLA improvements also extend to the Northern Hemisphere. These FOAM SLA enhancements in GOSI9 also lead to consistent improvements in zonal and meridional velocities relative to GO6 when compared to drifter velocity observations, which are not assimilated in FOAM. For instance, GOSI9 SLAs and velocities in FOAM ORCA12 are clearly enhanced in the western boundary currents, such as in the Brazil-Malvinas confluence, Gulf Stream and Kuroshio current, as well as along the ACC path.



For the sea ice, FOAM GOSI9 ORCA12 improvements are mixed when compared to GO6. The excessive FOAM GO6 sea ice melting over the Arctic summer, which was shown by Barbosa-Aguiar et al. (2024), is alleviated in GOSI9. This is likely because the SIC DA changes are proportionally propagated to more prognostic variables in SI$^3$ than in CICE, including the melt pond variables. Therefore, the ponding can be adjusted when new sea ice is introduced by DA, so that the pond albedo, which is lower than the ice albedo, does not lead to more sea ice melting in FOAM GOSI9 over the Arctic summer. Despite the Arctic summer improvements, FOAM GOSI9 SIC results are slightly worse than GO6 in the other Arctic seasons, as well as throughout the year in the Antarctic. Future work will aim at improving the background error covariances for SIC in GOSI9, as they are currently derived from a CICE run. There is also room to further improve the model tuning since the sea-ice model parameter settings of ORCA12 are the same as in ORCA025.

In addition to clear improvements on the global FOAM ORCA12 system, GOSI9 updates also have a large potential to enhance reanalysis runs, particularly in the pre-Argo period. The SLA, SST and T/S OmB statistics from a FOAM GOSI9 run withholding profile observations are significantly better than its equivalent GO6 run, particularly in the Southern Hemisphere. The heat content and AMOC transports are also more temporally consistent in FOAM GOSI9, drifting much less than in GO6 when T/S profile observations are not assimilated. This reinforces that the projection of the SLA DA signal onto large-scale temperature at depth, which no longer affects GOSI9, can significantly degrade the sub-surface temperatures in GO6 with negative impacts on the heat content and AMOC. The impact of this DA issue on Met Office reanalyses has already been reported in previous ocean reanalysis intercomparison studies. Jackson et al. (2019) showed that the Met Office reanalysis has a discrepant AMOC stream function near the equator when compared to other ocean reanalyses, whereas Mignac et al. (2018) decided not to use the Met Office reanalysis in their study because of its unrealistic AMOC transports in the South Atlantic. Thus, it is expected that GOSI9 results will lead to more potential for use of Met Office ocean reanalyses in climate studies.

Although the large-scale DA corrections for model temperatures have been removed in GOSI9, other work has been done to improve the specification of the background error correlations in FOAM. For instance, the implementation of a hybrid ensemble/variational assimilation scheme in NEMOVAR, as described in Lea et al. (2022), adds flow-dependent ensemble information to the background errors and improves the assimilation results when compared to the 3DVar scheme used here. Future work will implement this improved assimilation scheme in GOSI9 and evaluate the impact of using ensemble-derived, flow-dependent covariances for the background errors. Other avenues of future work to further improve GOSI9 results include the implementation of a T/S model bias correction scheme, as in Balmaseda et al. (2007) or Lellouche et al. (2018), to deal particularly with model biases in the pre-Argo period. Although ORCA12 DA is currently performed at ORCA025 resolution, the computational time for running NEMOVAR in GOSI9 has been greatly reduced with the implementation of the multi-resolution covariance modelling. Therefore, the next step for the global FOAM system is to test the impact of implementing the DA at ORCA12 full resolution to allow a better initialisation of features at mid- to high-latitudes where the correlation length-scales can be short, and around complex bathymetry not represented well on the ORCA025 grid.

Given that the ocean and sea-ice components of the different Met Office forecasting systems are initialised using the same DA framework, it is expected that not only FOAM but also the coupled NWP and GloSea seasonal forecasting systems will benefit



from the GOSI9 improvements shown here. The GOSI9 updates are planned to be implemented operationally in the next scientific upgrade of the Met Office operational suite (version 47), due in 2025, with a large potential to impact positively a wide range of end users across the different Met Office forecasting systems.

**Author contributions.** DM wrote the paper and ran the model experiments. DM, JW, DL, MM, JW and MP contributed with GOSI9 DA implementations. AW and AV contributed with NEMOVAR developments. CG, DS and DF contributed with NEMO updates in GOSI9. EB contributed with implementing the new sea ice model SI$^3$ as part of GOSI9. JB contributed with the AMOC diagnostics. KH, MB and RR provided expertise on data assimilation, ocean modelling and reanalysis results.

**Competing interests.** The authors declare no competing interests.

     **Code and data availability.** Details of how to download the NEMO and SI$^3$ used in GOSI9 can be found at https://doi.org/10.5281/zenodo.6334656 (Madec et al., 2022). The CICE5 (Hunke et al., 2015) used here in GSI8.1 is available from the Met Office code repository at https://code.metoffice.gov.uk/trac/cice/browser. Due to intellectual property copyright
restrictions, we cannot provide the source code of NEMOVAR.
Met Office Hadley Centre EN4 temperature and salinity quality-controlled profile observations were downloaded from https://www.metoffice.gov.uk /hadobs/en4/download-en4-2-1.html and assimilated in the simulations; Met Office Hadley Centre EN4 temperature objective analysis was downloaded from the same place for the heat content diagnostics. Global Drifter program data were downloaded from the Copernicus Marine Service, doi: 10.17882/86236. SSMIS data from Ocean
and Sea Ice Satellite Application Facility were obtained from https://osi-saf.eumetsat.int/products/sea-ice-products. AMOC rapid transports were downloaded from https://rapid.ac.uk/rapidmoc/rapid_data/datadl.php. This study has been conducted using E.U. Copernicus Marine Service Information: https://doi.org/10.48670/moi-00146 (altimetry data). Due to the size of FOAM runs, which require a large storage space of more than 3 TB, the full model output fields are not made available. They can be shared for non-commercial research use by contacting the authors.

     **Acknowledgements.** This work was funded partly through the Weather and Climate Science for Service Partnership (WCSSP) India, a collaborative initiative between the Met Office, supported by the UK Government's Newton Fund, and the Indian Ministry of Earth Sciences (MoES). We also acknowledge funding from the Royal Navy. Additionally, we have benefited from configurations developed by the Joint Marine Modelling Programme, a partnership between the Met Office, National
Oceanography Centre, British Antarctic Survey, and the Centre for Polar Observation and Modelling.



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
