# Peer review of "Improvements to the Met Office's global ocean-sea ice forecasting system including model and data assimilation changes"

_EGUsphere, 2024_

## Referee Comment (RC2)

**Review of: Updates to the Met Office's global ocean-sea ice forecasting system including model and data assimilation changes**

by Davi Mignac, Jennifer Waters, Daniel J. Lea, Matthew J. Martin, James While, Anthony T. Weaver, Arthur Vidard, Catherine Guiavarc'h, Dave Storkey, David Ford, Edward W. Blockley, Jonathan Baker, Keith Haines, Martin R. Price, Michael J. Bell, and Richard Renshaw

December 10, 2024

**Manuscript Synopsis**

The article is a system definition for the lastest version of the Met Office FOAM ocean data assimilation system. As such, this article is an important reference for testament of the current configuration of this system, and needs to be published. In general, the manuscript is well written and complete in its scientific rigour. I have only a few major comments that need to be addressed before publication. In particular, since the balance relationship scheme employed by NEMOVAR plays a critical role in the changes to the system, I believe a review of this scheme, previously described in the litereature, is warranted in preparation for its role in the changes. Although, for the most part, the changes and rationale for the changes are well written and easily followed with the evidence (figures) given, I did not find this as convincingly so for the addition of the Brunt-Väisälä criteria described in Section 3.2. The evidence (figure 4) is not as unambigiously positive as the authors would seem to suggest, and I feel a more detailed walk through this evidence to better point the reader to the benefit (or at least worst of the alternatives) that this change provides. A similar charge might also be leveled with regards to the benefits the updated version provides to the overturning circulation (latter portions of Section 4.3). Again the evidence (figure 14) fails to convince me of the positive impacts, at least not without further description – although in this case, at least part of the problem may be due to figure presentation deficiencies.

**My recommendation is Minor Revisions**

**Major Comments**

1. A large portion of the justification for removing the longer covariance lengthscale from the balanced portion of temperature background covariances – and therefore sea surface height and salinity increments – is based on the balance operators used in the NEMOVAR assimilation scheme. I believe a more detailed reminder of what those balance operators are beyond the stated "physically-based balance relationship" (l. 146) is required. The balance relationships of dynamical height adjustment and maintenance of neutral density, are critical to understanding why the longer temperature covariance lengthscale may be sub-optimal, but the contribution to an unbalanced salinity increment might have lesser negative effects. Although this might be obvious to the authors, it is sometimes necessary to lower the conversation to the readers potential ignorance. While the Waters et al. [2015] reference (BTW: An addition of the DOI to the reference would be useful: https://doi.org/10.1002/qj.2388) does extensively discuss this, I believe a more explicitly stated reminder would be useful here, possibly further referencing some of the earlier linearization work. Particular emphasis should be placed on why the introduction of a large scale covariance on temperature might bleed corrections to the unbalanced, barotropic sea surface anomaly (SLA) into the sub-surface temperature. By the way, the statement (l. 188) that "large-scale error covariances have a 400 km length-scale for temperature, unbalanced

salinity and unbalanced SSH" is correct [Mirouze et al., 2016], but it is also confusing! With the start of that sentence discussing the shorter length scale convariances, this statement somehow suggests to me that unbalanced SSH also has two convariances length scales, when in reality, it only has the longer, 400 km length scale.

2. I believe Figure S4 should be elevated to the main article. Discussion of why the SSH balance relationship is now applied throughout the mixed layer in GOSI9 (or why it was turned off through the mixed layer in GO6), is quite brief, and seems quite conclusively demonstated by Figure S4. Furthermore, it is equally, if not more, important, than the subsequent discussion of then mitigating some of the negative effects of that change (along with the removal of the long temperature covariance lengthscale) via the Brunt-Väisälä criteria in subsection 3.2. What is not expanded on, however, is whether the change to applying the balance throughout the mixed layer is related to the change to a single temperature covariance length scale. (I.e. I presume previous versions of the system better performed when the this relationship was not applied in the mixed layer.)

3. I found the rationale for applying the Brunt-Väisälä criteria to determine if the increment should be retained as less than convincing, at least with the current discussion and presentation of Figure 4. Showing the change of mean state at 1200m is a little confusing, as judging from the mean error shown in the profile of Figure 4d, it is not immediatedly obvious that the mean state of GOSI9+BV at that level is better than GO6 – and it is only just improving from being worse even then GOSI9 without BV slightly higher in the water column. I would think it might be better to highlight the mean temperature somewhere around 500m, where you can show that a more accurate mean state is achieved by **not** retaining the increment – and thus possibly increasing slightly the RMSE, although this latter part of the argument depends somewhat on whether the OMB statistics displayed are from the IAU phase (assumed), or the trial phase. This then can also lead, by not instigating spurious deep convection, to better RMSE and mean error lower down (below 1000 and 1300m respectively) in the column. I do not believe it is completely accurate to describe the mean state as being improved from 250m and below (l. 247), as it is fairly obviously degraded at 750m, at least relative to GO6. Finally, the Figure S6 is perhaps more definite in showing the improvement – although here the cost (increased RMSE) of not retaining the increment at certain levels is not as obvious. I would keep with the current discussion of the improvement, when well described, in the Labrador Sea deep convection region, but note to the reader further evidence for the Mediterranean overflow region is shown in the supplementary material.

4. Finally, I found the discussion of GOSI9 improving the overturning circulation (AMOC) in the latter portions of Section 4.3 (ll. 493 – 509) also not particularly convincing, at least by how it is served by Figure 14. The principle conclusion of section 4.3 in general, is that GOSI9 is more stable to the lack of sub-surface observations, or more specifically, mitigates the negative consequences of sea level anomaly assimilation in the absence of correcting sub-surface observations. The discussion on the AMOC attempts to then extend this improvement when lacking sub-surface profiles to the AMOC. This is convincingly demonstrated for the AMOC at 26.6°N (Figure 14a), where the divergence of the non-profile assimilation run with regards to the underlaying RAPID observations is convincingly shown. This is less convincingly shown at 50°N (Figure 14b) in the absence of underlying observations, which could be rectified by instead considering overturning along the OSNAP section, for which observations do exist, and the FOAM system has previously been evaluated [Jackson et al., 2019]. This is even less convincingly demonstrated by the the AMOC at 30°S, where the GOSI9, no T/S profile assimilation (cyan) line appears to be the outlier instead of GO6 no profile (orange), and appears to be going in the unwanted direction of too small, and even negative southern hemisphere AMOC that resulted in FOAM not being used in the Mignac et al. [2018] analysis. I would encourage the authors to at least include an AMOC estimate along the OSNAP section as previously shown in Jackson et al. [2019], with accompanying observational estimates, which should exist for the examined 2019.

5. Furthermore, I disagree with the implicit statement – it is not actually stated, but just somewhat implied by the discussion – that GOSI9 might improve (ll. 552-556) the "discrepant" equatorial overturning FOAM demonstated in GO6, which is only somewhat, and not conclusively, demonstrated through Figures 14d-h, or the "unrealistic" southern hemisphere AMOC suggested by Mignac et al. [2018], which is not demonstated at all in Figures 14d-h as the figure does not seem to go as far south

as 35°S in order to compare with Mignac et al. [2018]. Although as stated (ll. 501-503) some of this might await a proper evaluation of a FOAM reanalysis. .

**Minor Comments**

1. The manuscript mentions several assimilation method and ancillary input changes (ll. 71-74), along with the testing of new observational types (ll. 72-73). Most of the assimilation method and ancillary information changes have been implemented (or continue to be used) in GOSI9, but none of the mentioned new observation types are used – for the most part, due to the fact that they would not be operationally available. It might be interesting to return to this in the summary (i.e. could SSS assimilation [Martin et al., 2019] further allow removal of the longer salinity covariance scale; could U/V assimilation [Waters et al., 2024] improve equatorial transports, etc.).

2. **All** Figures: It is not mentioned whether the observation minus background statistics shown throughout the manuscript come from the trial run or from the IAU run. It is known FOAM calculates both, and it would be useful to know which of the two is shown throughout the manuscript – assuming it is not a mix of the two. This would help with the interpretation of figures such as Figure 4; it would be natural for RMSE of the IAU step to increase at levels where the increment is not being applied.

3. (ll. 230-232) The adjustment of melt pond variables to changes in SIC DA changes seems quite important to improvements/changes in sea ice for GOSI9 (ll. 393-403). It is also a rather ambiguous statement. Further details would be warranted/appreciated.

4. ll. 194-195, or more generally the whole discussion of 2 versus 1 covariance length scales (ll. 179-198). It would be useful to remind the reader why two covariance lengthscales was implemented, which was primarily to correct the near-surface drifts seen in the southern ocean (Figure 2e-f) and Mirouze et al. [2016].

    - In light of only near surface differences between the 1T-1S and 1T-2S covariance lengthscale schemes, could a near surface only (damped/cutoff with depth) long length scale S covariance work even better?
    - The only difference in T-profile statistics seems to be a very deep ($> 1500$m) increase in global RMSE of 1T-1S over 1T-2S (Figure 2a). This perhaps contradicts my statement immediately above. But more for the sake of curiosity than anything else, which region is responsible for this increase – as it is not the southern ocean.

5. ll. 332-333, Section 4.2. It likely should be mentioned that the SLA degradations found in coastal regions and enclosed seas are due to model changes (Figure S7c), although degradations in the central and eastern North Atlantic, or Mediterranean outflow regions would be due to data assimilation changes (Figure S7e).

    - The Figures 5c/d and S7e/f should be identical, but are not? Why? In particular, the degradations in the Mediterranean Outflow region in Figure 5d seem worse than those in Figure S7f.

6. The model changes include changes to the fundamental state variables of Temperature and Salinity. Therefore there are changes to the observation operators, and to the balance operators of NEMOVAR. Given that observation minus background (OmB) statistics are being used to show differences between the two versions:

    - Can OmB statistics be fairly compared? The GOSI9 model only SST changes seem universally degraded, especially in regions of large potential T vs conservative T differences (Figure S1).
    - Can changes to SLA OmB statistics possibly be attributed to changes in the dynamic height balance operator? Like in the coastal and enclosed sea regions?

7. Figures 6&7. A spatial map of OmB RMSE for depths between 300-700m might also be useful to visualize the improvements to the system upgrades. Note: Actual depth range was chosen somewhat arbitrarily – but differences spanning 500m depth seem to be relatively global in breadth.

8. In light of Mignac et al. [2022] I am curious as to why / disappointed that no evaluation of sea ice thickness was performed.

9. The no profile experiments answer the systems ability to operate in the pre-Argo period (1993-2005) with the presence of satellite altimeter and SST observations. However, it does not answer to the systems ability to operate in the absence of altimeter observations prior to 1993 – or a transition from no altimeter to altimeter observations in late 1992 (as for instance shown in Figure S3), particularly in the absence of correcting T/S profile measurements.

   - It may be necessary to caveat statements of the usefullness of the ocean reanalysis to a particular time period (1993 onward), as "GOSI9 will lead to more potential use of Met Office ocean reanalyses in climate studies" may have different envisioned time scales depending on the user (seasonal/decadal/century scale).
   - Nevertheless, it might be worth commenting on the possibile usefullness of this GOSI9 version of the Met Office FOAM system to long, century(?) scale reanalysis – or defer that discussion to a manuscript dedicated to an ocean reanalysis.

**Presentation Comments**

1. I do not find Figure 1 particulaly useful. A statement that the SLA observation error in GOSI9 is larger than that used in GO6, with the difference largely attributed to the use of 4cm measurement error might suffice. The figure itself could be demoted to the supplementary material. Personally, as mentioned above, I would rather see Figure S4 promoted to the manuscript.

2. Profile Figures. I believe it would be useful to have profile plots explicitly (in figure) labelled with a (largish) T or S (presumably in the white space normally found from mid-depth downward on the right side of most T and S profiles. This would rectify the necessity to refer to the caption all the time (although admittedly, those more familiar with the plots will immediately identify the differing shapes of T and S profiles).

3. Figure 2. I had difficulty, at least on the printed page, to differentiate the '2' and '1' in the labelling. You cannot improve my eyesight, but the labels could be comfortably enlarged somewhat. Improving even further beyond that might entail changing the shape of the font used 1T-2S (helvetica) may be more distinguishable than 1T-2S (computer modern).

4. Figure 3. One needs a Ph.D. in colour combinations to identify the regions. Caption note would seem suffice to warn the reader, but confusion will still remain. I have no suggestions, however.

5. Figure 14. Vertical span of transport strength is not sufficient to adequately distinquish the differing experiment lines. In general, the figure seems oddly arranged. I would think a 3 across set of time series above a 5 across set of Hovmüller diagrams would work better, allowing the vertical transport strength axis to be stretched to near the size of the horzontal time axis. For time series with observed data points, an accompanying set of correlation coefficients would be useful. I am aware such an $r$-coefficient would **only** distinquish the models ability to correctly follow the observed seasonal cycle – which in turn is largely Ekman wind driven (i.e. will largely be identical across all experiments, and the observation), but this may still be useful.

**References**

L. C. Jackson, C. Dubois, G. Forget, K. Haines, M. Harrison, D. Iovino, A. Köhl, D. Mignac, S. Masina, K. A. Peterson, C. G. Piecuch, C. D. Roberts, J. Robson, A. Storto, T. Toyoda, M. Valdivieso, C. Wilson, Y. Wang, and H. Zuo. The mean state and variability of the north atlantic circulation: A perspective from ocean reanalyses. *Journal of Geophysical Research: Oceans*, 124(12):9141–9170, 2019. doi: https://doi.org/10.1029/2019JC015210. URL https://agupubs.onlinelibrary.wiley.com/doi/abs/10.1029/2019JC015210.

Matthew J. Martin, Robert R. King, James While, and Ana Barbosa Aguiar. Assimilating satellite sea-surface salinity data from smos, aquarius and smap into a global ocean forecasting system. *Quarterly Journal of the Royal Meteorological Society*, 145(719):705–726, 2019. doi: https://doi.org/10.1002/qj.3461. URL https://rmets.onlinelibrary.wiley.com/doi/abs/10.1002/qj.3461.

D. Mignac, D. Ferreira, and K. Haines. South atlantic meridional transports from nemo-based simulations and reanalyses. *Ocean Science*, 14(1):53–68, 2018. doi: 10.5194/os-14-53-2018. URL https://os.copernicus.org/articles/14/53/2018/.

Davi Mignac, Matthew Martin, Emma Fiedler, Ed Blockley, and Nicolas Fournier. Improving the met office's forecast ocean assimilation model (foam) with the assimilation of satellite-derived sea-ice thickness data from cryosat-2 and smos in the arctic. *Quarterly Journal of the Royal Meteorological Society*, 148(744): 1144–1167, 2022. doi: https://doi.org/10.1002/qj.4252. URL https://rmets.onlinelibrary.wiley.com/doi/abs/10.1002/qj.4252.

I. Mirouze, E.W. Blockley, D.J. Lea, M.J. Martin, and M.J. Bell. A multiple length scale correlation operator with application to ocean data assimilation. *Submitted to Tellus*, 2016.

Jennifer Waters, Daniel J. Lea, Matthew J. Martin, Isabelle Mirouze, Anthony Weaver, and James While. Implementing a variational data assimilation system in an operational 1/4 degree global ocean model. *Quarterly Journal of the Royal Meteorological Society*, 141(687):333–349, 2015. doi: https://doi.org/10.1002/qj.2388. URL https://rmets.onlinelibrary.wiley.com/doi/abs/10.1002/qj.2388.

Jennifer Waters, Matthew J. Martin, Michael J. Bell, Robert R. King, Lucile Gaultier, Clément Ubelmann, Craig Donlon, and Simon Van Gennip. Assessing the potential impact of assimilating total surface current velocities in the met office's global ocean forecasting system. *Frontiers in Marine Science*, 11, 2024. ISSN 2296-7745. doi: 10.3389/fmars.2024.1383522. URL https://www.frontiersin.org/journals/marine-science/articles/10.3389/fmars.2024.1383522.

---

## Author Comment (AC1)

**Response to Reviewer 1**

We would like to thank reviewer 1 for suggesting minor corrections which will improve the overall quality of the paper. Responses to the reviewer's comments are highlighted in blue.

Beside minor corrections, I can support the manuscript for publication however I am concerned about the fact GMD strongly recommends the dissemination of the code. In the present case, the main improvements come from the DA code that is not free. Ocean Science can be a good alternative

We have previously submitted this paper to Ocean Science (OS). However, the OS editor believed that this paper would be a much better fit to GMD, due to all the technical aspects of the system that are presented in the paper. Therefore, we decided to withdraw our OS submission after the OS editor feedback and resubmit the paper to GMD.

**MINOR COMMENTS**

**1) GENERAL**

There is a sort of confusing nomenclature throughout the manuscript that I encourage the Authors to clarify for readability purposes. To my understanding, GO6/GOIS9 are generally used to point previous and new system while ORCA12 /ORCA025 are used to identify the resolution.

Yes, you are correct. GO6 and GOSI9 correspond to different versions of the global ocean and sea ice models used by the different Met Office forecasting systems, including FOAM. They are used in the manuscript to distinguish the old and new FOAM systems. ORCA12 and ORCA025 are used to identify the grid resolution of the sea ice and ocean models.

- In the Figures GO6 /GOI9 are used to differentiate the experiments, while the resolution (ORCA12 or ORCA025) is written in the caption. This is however confusing since ORCA12 parametrisation is different between GO6 and GOI9. I can suggest expanding the table 2 by introducing an different name for each experiment, something like GOIS9e12, GOIS9e025 etc.

We understand the reviewer's point here, however if we use the notation of "e12 and e025" to distinguish the different model resolutions, experiment names can be too long, for example GOSI9e025-NoTSProf. Furthermore, the two model resolutions are not directly compared in the paper. Section 4.2 only evaluates results from ORCA12 (which is the model resolution used to run FOAM forecasts operationally) and Section 4.3 only evaluates results from ORCA025 (which is the model resolution used to run FOAM reanalyses).

To make this point clearer in the paper, we included the model resolutions in the title of each result section. Section 4.2 has been changed to "Impacts on the global FOAM system **at 1/12º** with all observation types assimilated", whereas Section 4.3 has been changed to "Potential impacts on the pre-Argo reanalysis with the global FOAM system at **1/4º**".

- The use of ORCA12 DA is to be avoid since DA is performed at ¼° to my understanding. If two different DA set up at 1/4 ° are used, a description can be added to table 2.

We agree with reviewer and the use of ORCA12 DA has been removed from the paper.

- "FOAM" is used an attribute to both the system or the resolution, i.e. FOAM GO6 or FOAM ORCA12, etc., my impression is that you can safely remove "FOAM" everywhere.

We agree with the reviewer that this point can be improved. The term FOAM has been removed when it is associated to the model resolution. However, we will keep the term FOAM when it is associated to GO6 and GOSI9. Both GO6 and GOSI9 correspond to global ocean and sea ice model versions that are also used by other Met Office forecasting systems, such as the coupled NWP and seasonal forecasting system. Therefore, we believe it is important to reinforce throughout the paper that GO6 and GOSI9 results are associated with the old and new versions of the FOAM system.

**2) SPECIFIC**

Line 25: "In the absence of profile DA [..]". The sentence is difficult to follow and "profile DA" is not clear. Please rephrase it you can start with something like "Limited to the assimilation of surface data only [..]"

Done.

Line 96-101: I do not understand if those improvements are applied only to 1/12 ° or also to 1/4°

GOSI9 model changes are always applied to both ORCA12 and ORCA025 configurations. To make this point clear, the following statement has been added at the beginning of Section 2:

"GOSI9 model changes presented below are applied to both ORCA025 and ORCA12 configurations".

Figure F3 is blurry please replace it.

We believe you meant Figure S3 from the supplementary material. A new Figure S3 has been added.

Line 193 and Figure 2: Those results seem not conclusive, is there a reason why statistics are calculated only between Jan-May 2019 without covering the full year? Do you expect the results to be the same by including more statistics?

In the process of testing each individual GOSI9 change to the FOAM system in Section 3.1, we ended up running shorter experiments (varying from 1 to 6 months) which were enough to demonstrate the main impact of each change. Therefore, we could efficiently evaluate the impact of each GOSI9 change on FOAM and justify our choices before running the new FOAM system for 1 year with all changes together.

In our opinion, there is already a clear positive impact on the OmB temperature statistics at depth in the South Pacific and Southern Ocean from removing the long length-scale of temperature corrections, even though the runs only correspond to January-May 2019. This is even more evident in the OmB temperature statistics of 1-year GOSI9 runs with all changes

in Section 4.2 and 4.3, which validate our decision to remove the long length-scale temperature corrections from DA (as shown by Fig. 2).

Line 220: Is the minimization in GOIS9 always converging within 120 iterations? If not, can the Authors comment whether it is a noise problem, caused by the use of a first-order minimizer, or else.

In NEMOVAR, convergence depends on some convergence criteria. The one currently specified is not always met in FOAM with 120 iterations. However, we have run FOAM with NEMOVAR using different number of iterations: 40, 80, 120, 160 and 240. After 120 iterations, the OmB statistics are statistically indistinguishable between the runs. The use of a maximum number of iterations, rather than just waiting for the minimisation to converge, is done to ensure that the computational time of running NEMOVAR operationally is similar each day, so that we know we can produce the analysis and forecasts in time for our users.

Fig4: Similarly to my previous doubt, I am not sure the results in Fig4 are conclusive. There is a small difference among experiments and the statistics consider only 4 months. These results can be related to the specific position of few Argo that do not fully span the depths and the area. Can the Authors add the number of observations, extend the statistics or provide some study on the significativity?

We found a mistake with the land-ocean mask used to select the profiles and calculate the OmB temperature statistics for the Labrador Sea in Fig. 4d. We were not considering all the ocean area of the Labrador Sea but instead a much smaller region. This has been corrected and the positive impacts of using Brunt-Vaisala checks in the Labrador Sea are now larger at depth and convincing when compared to profile observations there.

We have plotted in Fig. 4a-c all the lat/long positions of the profile observations used to calculate the OmB temperature statistics between January and April 2019. We also added the number of profile observations in the caption of Fig. 4. The OmB statistics were calculated against 157 profiles which are well spread in the Labrador Sea.

Line 288: How many cores per node? Which is the frequency of the cpu?

We have added those details in the paper:

"Therefore, although FOAM GOSI9 has tripled the number of iterations, the DA at ORCA025 resolution takes 13 minutes in both FOAM GO6 and GOSI9 running on 15 computational nodes, each of which has 36 CPU cores on two 18 core Intel Broadwell Xeon 2.1 GHz processors."

Lines 310-315: Can the Authors specify whether there is some preprocessing on the drifter velocities? are the drifters without drogues, used?

Before being disseminated by the Copernicus Marine Service, the drifters are subjected to several quality control tests (Notarstefano et al., 2010; Elipot et al., 2016). The quality tests consist of applying drogue-loss detection algorithms and applying the wind stress and 10-m wind components from ECMWF weather forecasts at each drifter location. This determines consistency from model winds and calculated currents using spectral analysis and a regression analysis. The references above have been added to the paper, so the readers can further look at the details of the drifter velocity pre-processing methodology in case they would like to.

Drifters that are deemed to have lost their drogues are excluded from our assessment. Additionally, we have not accounted for the Stroke drift effects on the model currents. These details have also been added to the paper in Section 4.1.

**References:**

Elipot, S., Lumpkin, R., Perez, R. C., Lilly, J. M., Early, J. J., and Sykulski, A. M.: A global surface drifter data set at hourly resolution, J. Geophys. Res.-Oceans, 121, 2937–2966, https://doi.org/10.1002/2016JC011716, 2016.

Notarstefano, G., Gerin, R., Bussani, A., Bolzon, G., and Poulain, P. M.: Real Time Quality Control and Validation of Current Measurements Inferred from Drifter Data Within Copernicus in Situ TAC, CMEMS-INS-DRIFTER-RTQC, http://dx.doi.org/10.13155/74299, 2010.

325 "4.2 Impacts on the […] with all observation types assimilated": which is the full set of observation assimilated? It is written that it is similar to the operational system, but the latter may vary in time. For example it is not clearly whether the satellite SST is assimilated or not together with the drifter SST. Is the new rejection algorithm impacting the number of insitu profile assimilated significantly?

The full set of assimilated observations in this paper is the same as described in Tab. 1 of Barbosa-Aguiar et al. (2024). This is a very extensive table listing all the satellite and in situ products that we assimilate operationally. This is the reason why we refer the readers to this table in the first paragraph of Section 4.1, so we do not need to have the same long table in the paper. In this table we assimilate both satellite and in situ SST observations, although OmB SST statistics in the paper are only calculated against in situ (drifters) SSTs. We made this clear in Section 4.1:

"Additionally, even though OmB SST statistics are calculated only with respect to in situ drifters, swath SSTs are also assimilated from a variety of satellite sources (see Tab. 1 in Barbosa-Aguiar et al., 2024)."

The rejection of T/S increments is determined during the process of applying them to the model temperature and salinity, so that we end up rejecting increments at the model grid points (rather than specific observations). Since the rejection algorithm targets regions of water mass convection, only less than 2% of the model grid points have their vertical profile of T/S increments rejected on each assimilation cycle. This has been added to the first paragraph of Section 3.2.

**References:**

Barbosa Aguiar, A., Waters, J., Price, M., Inverarity, G., Pequignet, C., Maksymczuk, J., Smout-Day, K., Martin, M., Bell, M., King, R., While, J., and Siddorn, J.: The new Met Office global ocean forecast system at 1/12th degree resolution, Q.J. Royal Met. Soc., 1–26, https://doi.org/10.1002/qj.4798, 2024.

418: "Additionally, the same SI3 settings from […]": if I understood correctly the sea-ice DA is not changed passing from the old to the new system. Is it so? Is the same parametrization used for ORCA12 and ORCA025 in SI3?. What about CICE (old system)? Was there a different parametrization for ORCA12 and ORCA025?

The sea ice DA and how the SIC increments are applied to the model are the same in GO6 and GOSI9. To make this clear, we have changed the text in the last paragraph of Section 3.1:

"Although the sea ice model changed from CICE to SI$^3$ in GOSI9, the DA and how the total SIC increments are added to the sea ice model remain the same".

The sea ice model settings are the same between ORCA025 and ORCA12 in both CICE and SI³. To make this clear, we have changed the text in the last paragraph of Section 4.2:

"Additionally, the same sea ice model settings from ORCA025 are applied to ORCA12 in both CICE and SI³…"

421 "Potential impacts […] pre-Argo reanalysis": I have not understood whether satellite sst are assimilated in the pre-argo era configuration.

This has been made clear at the beginning of Section 4.3:

"Along-track SLA, satellite and in situ SST, and SIC data are still all assimilated in those experiments, since there are data available of those observation types during the pre-Argo period."

422-426: The Authors missed to comment the importance of a good initial condition in pre-argo era. Especially in the southern hemisphere, the model state can be far from true state and the assimilation of few SLA measurements can exacerbate any hidden biases. Initial condition in 2019 are instead well spin up and the absence of drastic changes in OHC can partially come from that. Please add a comment on this.

We have made this point clear in the paper when presenting the heat content results:

"It is also important to highlight that the initial condition of our experiments is well spun up, and one would expect the model to be much less constrained by observations in the pre-Argo period, particularly in the Southern Hemisphere. Although the heat content results are promising, future work will involve running longer FOAM GOSI9 reanalyses to look at the heat content drifts in the pre-Argo period."

430: "GOSI9-NoTSProf run […] SLA RMSDs in comparison to its original […]" I was expecting GOSI9-NoTSProf to be closer to SLA obs since the system is assimilating mainly SLA data. Why the RSME is similar to the one that ingest insitu data too?

The experiment results show that the assimilation of in situ and SLA data, when the horizontal temperature length-scales are updated, is done in a physically consistent way (through the multivariate balance relationships in NEMOVAR), so that the assimilation of in situ data does not degrade the performance of the model's SLA fields.

493-497: Which is the method used to estimate the AMOC? Are the Authors using the RAPID decomposition? Probably this is the cause of the difference. I noticed that the assimilation seems to degrade the correlation with RAPID timeseries, do you have any reason why?

We are using the model velocities across the whole section to calculate the AMOC maximum as opposed to following the RAPID methodology. There have been a few papers, particularly Danabasoglu et al. (2021), showing that the differences in the AMOC calculated from these methods are small. Therefore, we can reasonably say that the differences seen at 26ºN are real, especially between GO6-NoTsProf and the other experiments, rather than dependent on the method. We have added to the paper how we calculate the AMOC for the RAPID array and for the OSNAP array, which has been added to the comparison in Fig. 14.

We also found a mistake which explains the low correlation between the model runs and the RAPID array. We were erroneously considering the RAPID transports from **February 2019** to **January 2020** and this should have been from January 2019 to December 2019. After fixing this, the model transports are much better correlated when compared to the RAPID array.

**References:**

Danabasoglu, G., Castruccio, F. S., Small, R. J., Tomas, R., Frajka-Williams, E., and Lankhorst, M.: Revisiting AMOC transport estimates from observations and models, Geophysical Research Letters, 48, https://doi.org/10.1029/2021GL093045, 2021.

---

## Author Comment (AC2)

**Response to Reviewer 2**

We would like to thank K. Andrew Peterson for his thorough review, which will help to improve the overall quality of the paper. Responses to the reviewer's comments are highlighted in blue.

**Major Comments**

A large portion of the justification for removing the longer covariance length scale from the balanced portion of temperature background covariances – and therefore sea surface height and salinity increments – is based on the balance operators used in the NEMOVAR assimilation scheme. I believe a more detailed reminder of what those balance operators are beyond the stated "physically-based balance relationship" (l. 146) is required. The balance relationships of dynamical height adjustment and maintenance of neutral density, are critical to understanding why the longer temperature covariance length scale may be sub-optimal, but the contribution to an unbalanced salinity increment might have lesser negative effects. Although this might be obvious to the authors, it is sometimes necessary to lower the conversation to the readers potential ignorance. While the Waters et al. [2015] reference (BTW: An addition of the DOI to the reference would be useful: https://doi.org/10.1002/qj.2388) does extensively discuss this, I believe a more explicitly stated reminder would be useful here, possibly further referencing some of the earlier linearization work. Particular emphasis should be placed on why the introduction of a large scale covariance on temperature might bleed corrections to the unbalanced, barotropic sea surface anomaly (SLA) into the sub-surface temperature. By the way, the statement (l. 188) that "large-scale error covariances have a 400 km length-scale for temperature, unbalanced salinity and unbalanced SSH" is correct [Mirouze et al., 2016], but it is also confusing! With the start of that sentence discussing the shorter length scale covariances, this statement somehow suggests to me that unbalanced SSH also has two covariances length scales, when in reality, it only has the longer, 400 km length scale.

Our initial intention in the first paragraph of Section 3.1 was to give a brief context of the NEMOVAR balance operators, without giving too much detail about them, as we wanted to focus on the subsequent impacts of the DA changes. Therefore, we thought that having a proper reference, such as Waters et al. (2015), would be enough since the balance operators are properly described there. However, we agree with the reviewer that some of the GOSI9 DA impacts are related to the balance operators. Therefore, to accommodate the reviewer's comment without disrupting the flow of the paper around the new DA changes, we wrote a few more sentences about the NEMOVAR balance operators and cited previous works in addition to Waters et al. (2015). The introductory content of NEMOVAR and the description of the balance operators have been moved to the beginning of Section 3, so that Sections 3.1 and 3.2 are solely focussed on the impacts of the GOSI9 DA changes.

Regarding the confusing statement in Line 188, we have split it into two sentences. The first one is related to the short length-scales whereas the second is related to the long length-scales, so hopefully this will avoid ambiguities.

I believe Figure S4 should be elevated to the main article. Discussion of why the SSH balance relationship is now applied throughout the mixed layer in GOSI9 (or why it was turned off through the mixed layer in GO6), is quite brief, and seems quite conclusively demonstrated by Figure S4. Furthermore, it is equally, if not more, important, than the subsequent discussion of then mitigating some of the negative effects of that change (along with the removal of the long temperature covariance lengthscale) via the Brunt-Vaisala criteria in subsection 3.2. What is not expanded on, however, is whether the change to applying the balance throughout the

mixed layer is related to the change to a single temperature covariance length scale. (I.e. I presume previous versions of the system better performed when this relationship was not applied in the mixed layer.)

Both the Brunt-Vaisala criteria and the changes in the SSH balance are applied to mitigate localised numerical instabilities exacerbated by DA when the long temperature covariance length-scale is removed. We changed the text to make this point clearer in Section 4.1:

"This change reduces DA water column instabilities in areas of water mass convection, such as the Mediterranean Outflow region, **which are exacerbated by removing the long length-scale component of temperature background errors (see Section 3.2)**. Applying the SSH balance throughout the whole water column results in improved SLA statistics in the Mediterranean Outflow, particularly for ORCA12 (see Fig. S4)."

Regarding Section 3.2, we have made a mistake in Fig. 4d, selecting a much smaller area than the Labrador Sea to calculate the observation-minus-background (OmB) statistics. This has been fixed and Fig. 4d now shows a much larger, convincing impact of using the Brunt-Vaisala criteria to reject T/S increments in the Labrador Sea. As both Fig. 4 and Fig. S4 are mitigating the negative effects of not having the long temperature covariance length-scale, we prefer to keep Fig. 4 in the main article and leave Fig. S4 in the supplementary material. In our opinion, there is no need to have both in the main article.

I found the rationale for applying the Brunt-Vaisala criteria to determine if the increment should be retained as less than convincing, at least with the current discussion and presentation of Figure 4. Showing the change of mean state at 1200m is a little confusing, as judging from the mean error shown in the profile of Figure 4d, it is not immediatedly obvious that the mean state of GOSI9+BV at that level is better than GO6 – and it is only just improving from being worse even then GOSI9 without BV slightly higher in the water column. I would think it might be better to highlight the mean temperature somewhere around 500m, where you can show that a more accurate mean state is achieved by not retaining the increment – and thus possibly increasing slightly the RMSE, although this latter part of the argument depends somewhat on whether the OMB statistics displayed are from the IAU phase (assumed), or the trial phase. This then can also lead, by not instigating spurious deep convection, to better RMSE and mean error lower down (below 1000 and 1300m respectively) in the column. I do not believe it is completely accurate to describe the mean state as being improved from 250m and below (l. 247), as it is fairly obviously degraded at 750m, at least relative to GO6. Finally, the Figure S6 is perhaps more definite in showing the improvement – although here the cost (increased RMSE) of not retaining the increment at certain levels is not as obvious. I would keep with the current discussion of the improvement, when well described, in the Labrador Sea deep convection region, but note to the reader further evidence for the Mediterranean overflow region is shown in the supplementary material.

As mentioned previously, the Labrador Sea area used for selecting the T profile observations in the OmB statistics was much smaller than it should be. Fixing this improves considerably the results of the OmB temperature statistics in the Labrador Sea, as shown by the corrected Fig 4d. The benefits of using the Brunt-Vaisala criteria are now clear in the Labrador Sea, with consistent impacts in the 500-2000 m mean differences and RMSDs. This is also consistent with showing temperatures at 1200 m in Fig. 4a-c, since the impacts of applying the Brunt-Vaisala criteria are clearly larger at depth in the corrected Fig. 4d. Fig. 4a-c also had their colourmap range decreased, making it easier to see temperature differences at 1200 m between the experiments.

The OmB statistics are from a one-day model forecast before the increments for that cycle have been included. In Section 4.1, we believe the statement below clarifies that:

"Although these observations are compared to the model background, **i.e. before being assimilated**, they cannot be treated as fully independent datasets".

Finally, I found the discussion of GOSI9 improving the overturning circulation (AMOC) in the latter portions of Section 4.3 (ll. 493 – 509) also not particularly convincing, at least by how it is served by Figure 14. The principle conclusion of section 4.3 in general, is that GOSI9 is more stable to the lack of sub-surface observations, or more specifically, mitigates the negative consequences of sea level anomaly assimilation in the absence of correcting sub-surface observations. The discussion on the AMOC attempts to then extend this improvement when lacking sub-surface profiles to the AMOC. This is convincingly demonstrated for the AMOC at 26.6°N (Figure 14a), where the divergence of the non-profile assimilation run with regards to the underlaying RAPID observations is convincingly shown. This is less convincingly shown at 50°N (Figure 14b) in the absence of underlying observations, which could be rectified by instead considering overturning along the OSNAP section, for which observations do exist, and the FOAM system has previously been evaluated [Jackson et al., 2019]. This is even less convincingly demonstrated by the the AMOC at 30°S, where the GOSI9, no T/S profile assimilation (cyan) line appears to be the outlier instead of GO6 no profile (orange), and appears to be going in the unwanted direction of too small, and even negative southern hemisphere AMOC that resulted in FOAM not being used in the Mignac et al. [2018] analysis. I would encourage the authors to at least include an AMOC estimate along the OSNAP section as previously shown in Jackson et al. [2019], with accompanying observational estimates, which should exist for the examined 2019

We have now included the comparison between the model and the OSNAP transports in Fig. 14b. It is very consistent with the comparison with the RAPID transports, showing that GO6-NoTSProf is clearly overestimating the transports along these two arrays, while GOSI9-NoTSProf follows very closely the observed transports. This now demonstrates more convincingly the GOSI9 impacts on the ocean circulation, especially in the North Atlantic.

We agree with the reviewer that GOSI9-NoTSProf shows weaker transports at 30S in June-October 2019, but they seem to recover in November and December, being similar to the other runs. This is why we state in the paper that a long reanalysis should be run in the future, so we will be able to conclusively evaluate the GOSI9 impacts on the South Atlantic MOC.

Furthermore, I disagree with the implicit statement – it is not actually stated, but just somewhat implied by the discussion – that GOSI9 might improve (ll. 552-556) the "discrepant" equatorial overturning FOAM demonstrated in GO6, which is only somewhat, and not conclusively, demonstrated through Figures 14d-h, or the "unrealistic" southern hemisphere AMOC suggested by Mignac et al. [2018], which is not demonstrated at all in Figures 14d-h as the figure does not seem to go as far south 2 as 35°S in order to compare with Mignac et al. [2018]. Although as stated (ll. 501-503) some of this might await a proper evaluation of a FOAM reanalysis.

In Fig. 2 of Jackson et al. (2019), the AMOC stream function of the Met Office reanalysis is compared to several other reanalyses. It is clear that the Met Office reanalysis has a much stronger equatorial transport, including discontinuities in the AMOC stream function, when compared to other reanalyses. This is a very similar pattern to what has been shown in GO6-NoTSProf with too strong equatorial transports (above 40 Sv) and AMOC discontinuities near the equator. Both aspects have been improved in GOSI9-NoTSProf, with AMOC transports decreasing by ~15 Sv in the equatorial region relative to GO6-NoTsProf, which is a clear

improvement in the equatorial transports. This has been added to the paper when discussing Fig. 14.

Mignac et al. (2018) and Jackson et al. (2019) raised similar concerns about the South Atlantic transports in the Met Office reanalysis. Jackson et al. (2019) stated the following:

"In particular, GloSea5 is suspect in the South Atlantic and near the equator (where there is a discontinuity in stream function strength): This issue has been traced to the method of assimilating sea surface height and will be the subject of a future publication".

Although we agree with the reviewer that South Atlantic MOC improvements are still inconclusive, this paper addresses the concerns raised by Jackson et al. (2019). These concerns are related to how our sea surface height assimilation impacts the ocean circulation, which has been improved in GOSI9. A GOSI9 reanalysis from 1993 to present is currently being run and a future publication will be able to show conclusive results in the South Atlantic circulation.

**Minor Comments**

The manuscript mentions several assimilation method and ancillary input changes (ll. 71-74), along with the testing of new observational types (ll. 72-73). Most of the assimilation method and ancillary information changes have been implemented (or continue to be used) in GOSI9, but none of the mentioned new observation types are used – for the most part, due to the fact that they would not be operationally available. It might be interesting to return to this in the summary (i.e. could SSS assimilation [Martin et al., 2019] further allow removal of the longer salinity covariance scale; could U/V assimilation [Waters et al., 2024] improve equatorial transports, etc.).

This is a good point. We have added the following sentence in the summary:

"Additionally, the potential of assimilating new observations in future operational versions of FOAM, such as the satellite sea surface salinity (Martin et al., 2019) and surface current velocities (Waters et al., 2024), may further improve near-surface salinity fields and equatorial transports in GOSI9, respectively."

All Figures: It is not mentioned whether the observation minus background statistics shown throughout the manuscript come from the trial run or from the IAU run. It is known FOAM calculates both, and it would be useful to know which of the two is shown throughout the manuscript – assuming it is not a mix of the two. This would help with the interpretation of figures such as Figure 4; it would be natural for RMSE of the IAU step to increase at levels where the increment is not being applied.

This has already been addressed in a previous comment. However, we reinforce here that all OmB statistics shown in the paper are from a one-day model forecast before the increments for that cycle have been included.

(ll. 230-232) The adjustment of melt pond variables to changes in SIC DA changes seems quite important to improvements/changes in sea ice for GOSI9 (ll. 393-403). It is also a rather ambiguous statement. Further details would be warranted/appreciated.

The increments are added to the sea ice concentration on each IAU step, and the sea ice volume is therefore updated by multiplying the new sea ice concentration, containing the increments, and the sea ice thickness. After this is done, we compute the changes in sea ice concentration and volume as a ratio (division) between the updated variables (after increments are added) and old variables (before increments are added). These changes in sea

concentration and volume are used to proportionally adjust the other prognostic variables. For variables based on the volume (e.g. snow volume, ice enthalpy, melt pond volume, etc), they are multiplied by the changes in the sea ice volume, whereas for the variables based on the area (e.g. melt pond area, etc), they are multiplied by the changes in the sea ice concentration. For the melt ponds variables specifically, this proportional adjustment is done with the opposite sign, so that when DA adds ice over summer, the ponding can be reversed, avoiding the feedback issues between the DA and the melt ponds (as seen in GO6).

We have now provided more detail on some of these aspects in Section 3.1, particularly about the melt pond adjustment.

ll. 194-195, or more generally the whole discussion of 2 versus 1 covariance length scales (ll. 179-198). It would be useful to remind the reader why two covariance lengthscales was implemented, which was primarily to correct the near-surface drifts seen in the southern ocean (Figure 2e-f) and Mirouze et al. [2016].

This reminder has been added to the paper:

"The large-scale error covariances have a 400 km length-scale for temperature, unbalanced salinity and unbalanced SSH, which are demonstrated to correct near-surface drifts, particularly for salinity in the Southern Ocean (see Mirouze et al., 2016)."

• In light of only near surface differences between the 1T-1S and 1T-2S covariance lengthscale schemes, could a near surface only (damped/cutoff with depth) long length scale S covariance work even better?

There is ongoing work to address this point. We are currently testing changes to the horizontal salinity length-scales by reducing the effect of the long length-scale below the surface layers which is showing some promise. If successful, this will be included in a future update to the FOAM system but is not being implemented as part of the changes described in this paper.

• The only difference in T-profile statistics seems to be a very deep (> 1500m) increase in global RMSE of 1T-1S over 1T-2S (Figure 2a). This perhaps contradicts my statement immediately above. But more for the sake of curiosity than anything else, which region is responsible for this increase – as it is not the southern ocean.

The global RMSE increase at depths greater than 1500 m with 1T-1S configuration is coming from the Labrador Sea. When testing the length-scale setups, we found that the Labrador Sea is very sensitive to how T/S corrections are applied horizontally. Having the 1T-1S configuration gives the worst results when compared to 1T-2S or 2T-2S. Even the 2T-2S configuration produces numerical instabilities and erroneously triggers deep convection, clearly degrading the OmB statistics when compared to the run applying the Brunt-Vaisala rejection algorithm (see corrected Fig. 4d in the paper).

ll. 332-333, Section 4.2. It likely should be mentioned that the SLA degradations found in coastal regions and enclosed seas are due to model changes (Figure S7c), although degradations in the central and eastern North Atlantic, or Mediterranean outflow regions would be due to data assimilation changes (Figure S7e).

This has been added to the paper in Section 4.2:

"We also note that the SLA degradations found in some coastal regions and enclosed seas in Fig. 5 are due to model changes, whereas SLA degradations in the central and eastern North Atlantic are due to DA changes (see Fig. S7)."

• The Figures 5c/d and S7e/f should be identical, but are not? Why? In particular, the degradations in the Mediterranean Outflow region in Figure 5d seem worse than those in Figure S7f.

We believe that the reviewer missed the statement in Section S7-S9 of the supplementary material:

"It is important to highlight that the impacts of the model and DA changes on FOAM GOSI9 were evaluated in ORCA025 only, as the ORCA12 configuration is quite expensive to run for one year. However, since both configurations show very similar impacts on the observation-minus-background (OmB) statistics when comparing FOAM GOSI9 against FOAM GO6 (see Section 4 in the paper), the results below should represent a valid evaluation of the model and DA update contributions to FOAM GOSI9 improvements."

The reason for small differences between Fig. 5c-d and Fig. S7e-f is that the breakdown of the model and DA impacts on GOSI9 was done with ORCA025 instead of ORCA12. The reason for this is that it is quite expensive to run 1-year experiments with ORCA12.

The model changes include changes to the fundamental state variables of Temperature and Salinity. Therefore, there are changes to the observation operators, and to the balance operators of NEMOVAR. Given that observation minus background (OmB) statistics are being used to show differences between the two versions:

• Can OmB statistics be fairly compared? The GOSI9 model only SST changes seem universally degraded, especially in regions of large potential T vs conservative T differences (Figure S1).

As stated in Section 4.1 of the paper, we believe the OmB statistics can be fairly compared:

"It is also worth noting that the temperature and salinity RMSD results for FOAM GO6 and GOSI9 are calculated from the EOS80 and TEOS10 variables, respectively. The magnitude of the errors is expected to be consistent whether using TEOS10 or EOS80. We investigated the impact of converting between absolute and practical salinity on the OmB values and found that it has a very small impact of the order of 0.001, which is much smaller than the salinity differences and RMSDs between FOAM GO6 and GOSI9 presented here."

With respect to GOSI9 OmB SST statistics being universally degraded due to model changes, we respectfully disagree. The RMSE percentage changes are quite small and mostly negligible, except for the Arctic region, where the model changes do clearly degrade the SST statistics. However, as seen in Fig. S1, the difference between potential and conservative temperature in the Arctic is much smaller when compared to other regions, such as the subtropical and equatorial regions.

• Can changes to SLA OmB statistics possibly be attributed to changes in the dynamic height balance operator? Like in the coastal and enclosed sea regions?

The inclusion of the mixed layer in the dynamic height balance is not the main reason for improvements in the SLA OmB statistics. It has a large impact on the Mediterranean Outflow SLA statistics in GOSI9, but it is still a very localised improvement. The negative impacts on the SLA OmB statistics in the coastal regions and enclosed seas are mainly due to model rather than DA changes (see Fig. S7).

Figures 6&7. A spatial map of OmB RMSE for depths between 300-700m might also be useful to visualize the improvements to the system upgrades. Note: Actual depth range was chosen

somewhat arbitrarily – but differences spanning 500m depth seem to be relatively global in breadth.

We do not think these additional figures are needed. We did calculate OmB profile statistics regionally, so we could consistently compare the statistics between GO6 and GOSI9 throughout the water column (up to 2000 m) and for different ocean regions. This convincingly shows the differences at depth between GO6 and GOSI9.

In light of Mignac et al. [2022] I am curious as to why / disappointed that no evaluation of sea ice thickness was performed.

We still do not assimilate sea ice thickness operationally. Additionally, although they are different sea ice models, CICE and SI$^3$ have largely the same sea ice physics. Therefore, small differences in the sea ice thickness are expected between GO6 and GOSI9. For this reason, we did not include any sea ice thickness comparison in the paper and focussed only on the evaluation of the sea ice concentration.

The no profile experiments answer the systems ability to operate in the pre-Argo period (1993-2005) with the presence of satellite altimeter and SST observations. However, it does not answer to the systems ability to operate in the absence of altimeter observations prior to 1993 – or a transition from no altimeter to altimeter observations in late 1992 (as for instance shown in Figure S3), particularly in the absence of correcting T/S profile measurements.

> • It may be necessary to caveat statements of the usefulness of the ocean reanalysis to a particular time period (1993 onward), as "GOSI9 will lead to more potential use of Met Office ocean reanalyses in climate studies" may have different envisioned time scales depending on the user (seasonal/decadal/century scale).

Done.

"Thus, it is expected that GOSI9 results will lead to more potential for use of Met Office ocean reanalyses in climate studies, particularly for the satellite altimetry era from 1993 onwards."

> • Nevertheless, it might be worth commenting on the possible usefulness of this GOSI9 version of the Met Office FOAM system to long, century(?) scale reanalysis – or defer that discussion to a manuscript dedicated to an ocean reanalysis.

We believe that this is beyond the scope of the paper where we focus on the impact of the changes during the altimeter period.

**Presentation comments**

I do not find Figure 1 particularly useful. A statement that the SLA observation error in GOSI9 is larger than that used in GO6, with the difference largely attributed to the use of 4cm measurement error might suffice. The figure itself could be demoted to the supplementary material. Personally, as mentioned above, I would rather see Figure S4 promoted to the manuscript.

We think Fig. 1 is useful as it shows a clear contrast between the magnitude of the SLA observation errors in GO6 and GOSI9. Since SLA observation errors in GO6 are much smaller than in GOSI9, this also suggests that the SLA assimilation in GO6 may be overfitting the observations, exacerbating the negative impacts of the SLA assimilation on the heat content and ocean circulation when T/S profile observations are withheld. Therefore, we prefer to keep Fig. 1 in the main article.

Profile Figures. I believe it would be useful to have profile plots explicitly (in figure) labelled with a (largish) T or S (presumably in the white space normally found from mid-depth downward on the right side of most T and S profiles. This would rectify the necessity to refer to the caption all the time (although admittedly, those more familiar with the plots will immediately identify the differing shapes of T and S profiles).

Done.

Figure 2. I had difficultly, at least on the printed page, to differentiate the '2' and '1' in the labelling. You cannot improve my eyesight, but the labels could be comfortably enlarged somewhat. Improving even further beyond that might entail changing the shape of the font used 1T-2S (helvetica) may be more distinguishable than 1T-2S (computer modern).

Done.

Figure 3. One needs a Ph.D. in colour combinations to identify the regions. Caption note would seem suffice to warn the reader, but confusion will still remain. I have no suggestions, however.

This is a tricky one. We tried different things, such as only having the coloured borders with no filling, but the figure was still polluted. After a few attempts of changing this figure, we concluded that the way this figure is presented in the paper is the best way when compared to the other alternatives that we tried.

Figure 14. Vertical span of transport strength is not sufficient to adequately distinquish the differing experiment lines. In general, the figure seems oddly arranged. I would think a 3 across set of time series above a 5 across set of Hovm¨uller diagrams would work better, allowing the vertical transport strength axis to be stretched to near the size of the horzontal time axis. For time series with observed data points, an accompanying set of correlation coefficients would be useful. I am aware such an rcoefficient would only distinquish the models ability to correctly follow the observed seasonal cycle – which in turn is largely Ekman wind driven (i.e. will largely be identical across all experiments, and the observation), but this may still be useful.

We tried to rearrange Fig. 14 as suggested by the reviewer, but we still prefer how it is presented in the paper. When rearranging the figure to the format suggested by the reviewer, the AMOC stream function plots become too small, as they consist of a row of 5 plots in this format. Therefore, our preference is to keep Fig. 14 as it is.

Regarding the correlation coefficients, we do not believe this is relevant to Fig. 14. The main goal of Fig. 14 is to show the GOSI9 impacts on the transport magnitudes and how they compare to observed transports. Furthermore, it is not ideal to calculate correlation coefficients based only on 12 numbers. Refined metrics about the GOSI9 impacts on the AMOC are planned to be shown in a future publication with a long GOSI9 reanalysis.

---

## Author Response (AR2)

**Response to Reviewer 1**

We would like to thank Reviewer 1 for reviewing the paper for a second time. Responses to the reviewer's comments are highlighted in blue.

The importance of present Figure 2 is still not clear to me. If I understood correctly, the removal of long-range temperature length-scale mitigates the problem with SLA assimilation, moreover it improves the BIAS and RMSE specifically in the Southern Ocean. The behaviour is opposite for salinity, the long-range length-scale can be kept since it does not enter the density calculation, moreover it improves the BIAS and RMSE especially in the Southern Ocean.

You understood correctly and your point above is what Figure 2 is intended to highlight, i.e. that removing the long temperature length-scales will improve the sub-surface temperatures, particularly in the Southern Ocean, by avoiding an inconsistent propagation of the SLA assimilation signal onto the large-scale sub-surface temperatures. Therefore, both RMSE and temperature biases are improved, particularly in the Southern Ocean.

I found that this opposite behaviour represents an interesting point to discuss in some more details since previous papers on FOAM system stated that the use of the sole short-range length-scale goes in the direction of improving the RMSE. The inclusion of long length-scale is meant not to reduce the RMSE but rather to reduce the presence of noisy fields in the case of sparse data, and to constrain the bias of tracer fields when observations are sparse (Blockley et al. 2014, Waters et al, 2013,2014).

The key paper to assess the impact of the NEMOVAR multiple length-scale approach is Mirouze et al. (2016). There is the following statement in the abstract taken from Mirouze et al. (2016):

"The multiple length scale operator has been implemented in NEMOVAR, a variational ocean data assimilation system. A dual length scale formulation was tested in a one-year reanalysis and compared with a single length scale formulation. The results emphasise the importance of estimating with great care the factors used within the combination. They also demonstrate the potential of the dual length scale formulation, in particular through a decrease of the innovation statistics for salinity profiles. The dual length scale formulation is now operational at the Met Office."

There is also this statement in the conclusion of Mirouze et al. (2016):

"When compared with a single length scale formulation, the results show an increase of the RMS error for innovations where a dense observation network is assimilated (SST and SSH)."

In summary, this work found that using a long length-scale for salinity can be beneficial. However, for other variables this may not be true and potentially lead to an increase in RMS errors as well. We believe the findings of our paper are consistent with Mirouze et al. (2016). We have changed the text in our paper to better highlight the aspects of using long length-scales in NEMOVAR:

"These large-scale error covariances are demonstrated to correct near-surface drifts, particularly for salinity in the Southern Ocean, however their impact on other variables, such as SST and SSH, is not clear and might lead to degradations in areas where a dense observation network is assimilated (see Mirouze et al., 2016)."

**Reference:**
Mirouze, I., Blockley, E. W., Lea, D. J., Martin, M. J., and Bell, M. J.: A multiple length scale correlation operator for ocean data assimilation, Tellus A: Dynamic Meteorology and Oceanography, 68, https://doi.org/10.3402/tellusa.v68.29744, 2016.

Based on the results of Figure 2, are the Authors confident with this opposite behaviour in temperature and salinity, over a longer period says one or few years?

The use of only a short length-scale for temperature and both short and long length-scales for salinity is tested for 1 year as part of the GOSI9 changes in Section 4.2 and Section 4.3. Therefore, we are confident about this change for a timeline of 1 year. Going beyond 1 year, we are currently running a GOSI9 reanalysis to confirm the 1-year results of this paper. We made a statement to reflect this point in the paper:

"Although the positive impacts of GOSI9 changes are shown here only for one-year experiments, we expect that GOSI9 results will lead to more potential for use of Met Office ocean reanalyses in climate studies, particularly for the satellite altimetry era from 1993 onwards. Therefore, future work will involve running a GOSI9 reanalysis for the satellite altimetry era."

If not, I can suggest to remove Figure 2. My impression is that it lacks information that confirm the significativity of such results within 5 months: are such OmB statistics stable in time or do the improvements come only from a short period within the 5 months? Are the number of observations, used to evaluate OmB, similar in the three experiments? A further check on the number of observations can be beneficial, since two mistakes have been already rectified in other figures.

The OmB RMSD for temperature of the 1T-2S experiment improves over the first 5 months, particularly between 500-1500 m, for both the Global and Southern Ocean. As we keep the long salinity length-scale in both 1T-2S and 2T-2S, the impact on the OmB RMSD for salinity is less clear over time, however more consistent improvements are seen towards the end of the first 5 months for both the Global and Southern Ocean. The Hovmöller plots below show the OmB RMSD percentage improvements (degradations) in blue (red) of 1T-2S relative to 2T-2S for temperature and salinity in both the Global and Southern Ocean.

[Figure]

Figure: RMSD percentage improvements (degradations) in blue (red) of experiment 1T-2S relative to experiment 2T-2S between January and May 2019 when both are compared to profile observations.

We also plotted the OmB statistics for temperature and salinity as in Figure 2 of the paper but for specific months over the 5-month period, such as January, March and May 2019. This is consistent with the Hovmöller showing that OmB statistics do improve over time, particularly in southern latitudes.

[Figure]

Figure: Temperature (left) and Salinity (right) OmB statistics calculated for three specific months (January, March and May 2019) against profile observations in the Global Ocean and South Pacific for 2T-2S (red) and 1T-2S (blue) experiments.

We confirm that the number of observations used to produce the OmB statistics in Figure 2 of the paper (and in the figures above) is the same between the experiments.

**Response to Reviewer 2**

We would like to thank K. Andrew Peterson for reviewing the paper for a second time. Responses to the reviewer's comments are highlighted in blue.

L. 320: Validation experiment runs are initialized from a 2017-2018 run of GO6. Exact phrasing "They come from 2017-2018 ORCA12 and ORCA025 runs with the FOAM GO6 configuration (see Barbosa-Aguiar et al., 2024)." This is ambiguous. I believe what you meant is "They are initialized from the end of 2017-2018 ORCA12 and ORCA025 runs with the FOAM GO6 configuration on 1 Jan, 2019 (correct date?).

Done.

L. 325. Presented OmB statistics are calculated from ..., SST from in situ drifters, ... This is an accurate statement, but neglects to specify you are ignoring further OmB statistics from assimilated (but biased) satellite SST observations. Perhaps this is worth stating?

Changed as below:

"In addition to drifters, swath SSTs are also assimilated from a variety of satellite sources (see Tab. 1 in Barbosa-Aguiar et al., 2024). However, satellite SSTs can be biased and therefore are bias-corrected in NEMOVAR (While and Martin, 2019). For this reason, OmB SST statistics are calculated here only with respect to in situ drifters."

All OmB spatial plots: There is no warning/admission that zero OmB difference and no data are both signified by white colour mapping. In general this is not a problem (most observed areas have at least some non-zero shading), or as in the velocity plots, the rms error plots display the observation error holes.

We appreciate the comment, but we believe this is an extremely minor detail and the OmB spatial maps are clear enough to show the positive impacts of GOSI9.

Again OmB spatial plots: Presumably the OmB statistics have been binned into lat-lon bins or onto the model grid? The bin size is never stated, but looks relatively high resolution. Please state if possible in Figure 5 (assuming it remains the same in all figures).

Done. This information has been added to the paper in Figs. 5, 8, 9 and 10.

LL 462-464: However, it is worth highlighting that FOAM SLA statistics in the central and eastern North Atlantic are slightly better in GOSI9-NoTSProf (Fig. 10g) than in GOSI9 (Fig. 10e). This suggests that there could still be minor issues in assimilating SLA and T/S profile data together, even after the substantial SLA improvements caused by GOSI9 DA changes.

This also appears to be true in the equatorial Pacific -- and possibly the whole of the Pacific, just not degradations from GO6 (deeper blue improvements). Were velocity statistics also performed on the no profile runs to further support SLA OmB statistics. Not suggesting to add plots -- just possibly state velocity statistics confirm better enhanced adjustment to SLA innovations without (potentially) conflicting T/S innovations.

The text was changed as below:

"However, it is worth highlighting that FOAM SLA statistics in the central and eastern North Atlantic are slightly better in GOSI9-NoTSProf (Fig. 10g) than in GOSI9 (Fig. 10e), **as well as in the Pacific.**"

We did not calculate the velocity statistics for the no profile runs. We rather focussed on providing the heat content and AMOC diagnostics in Section 4.3 since they are key indicators from a reanalysis perspective.